# High-throughput mutagenesis reveals unique structural features of human ADAR1

SeHee Park[1], Erin E. Doherty [1], Yixuan Xie[1], Anil K. Padyana [2], Fang Fang[3], Yue Zhang[1], Agya Karki[1], Carlito B. Lebrilla [1,4], Justin B. Siegel[1,4,5] & Peter A. Beal [1✉]

Adenosine Deaminases that act on RNA (ADARs) are enzymes that catalyze adenosine to inosine conversion in dsRNA, a common form of RNA editing. Mutations in the human *ADAR1* gene are known to cause disease and recent studies have identified ADAR1 as a potential therapeutic target for a subset of cancers. However, efforts to define the mechanistic effects for disease associated ADAR1 mutations and the rational design of ADAR1 inhibitors are limited by a lack of structural information. Here, we describe the combination of high throughput mutagenesis screening studies, biochemical characterization and Rosetta-based structure modeling to identify unique features of ADAR1. Importantly, these studies reveal a previously unknown zinc-binding site on the surface of the ADAR1 deaminase domain which is important for ADAR1 editing activity. Furthermore, we present structural models that explain known properties of this enzyme and make predictions about the role of specific residues in a surface loop unique to ADAR1.

[1] Department of Chemistry, University of California, Davis, Davis, CA, USA. [2] Agios Pharmaceuticals, Cambridge, MA, USA. [3] Viva Biotech Ltd., Shanghai, China. [4] Department of Biochemistry and Molecular Medicine, University of California, Davis, Davis, CA, USA. [5] Genome Center, University of California Davis, Davis, CA, USA. ✉email: pabeal@ucdavis.edu

ADARs (adenosine deaminases that act on RNA) catalyze hydrolytic deamination of adenosine (A) in RNA generating inosine (I)[1–3]. Because this reaction changes the Watson-Crick hydrogen bonding specificity of the base, the consequences of A-to-I modifications on RNA function are wide-ranging and include changing the meaning of specific codons (recoding), redirecting splicing, altering miRNA recognition sites, and modulation of duplex RNA stability[4–6]. The different enzymes known to carry out the A-to-I conversion in humans are ADAR1 (p150 and p110) and ADAR2[7]. ADAR activity is required for nervous system function and altered editing has been linked to neurological disorders[8–10]. Furthermore, ADAR1 plays an important role in innate immunity[11–13]. Mutations in the *ADAR1* gene are known to cause the autoimmune disease Aicardi-Goutieres Syndrome (AGS) and the skin disorder dyschromatosis symmetrica hereditaria (DSH)[14–16]. In addition, ADAR1 mediated A-to-I editing of dsRNAs prevents over-activation of dsRNA sensing pathways, thus avoiding an auto inflammatory response[11,12]. In fact, a recent study has shown that ADAR1 knockout sensitizes certain types of tumors to immunotherapy by activating dsRNA sensing pathways in tumor cells, which in turn leads to an innate immune response[17]. Thus, investigating how ADAR1 activity is regulated and how ADAR1 differentiates self RNAs from non-self RNAs is crucial to understand the immune response pathways regulated by ADAR1. In addition, recent studies show the loss of ADAR1 function is lethal in a specific subset of cancers that display an interferon-stimulated gene signature identifying ADAR1 as a potential cancer therapeutic target[13,18–20].

The ADAR proteins have a modular structure with double-stranded RNA-binding domains (dsRBDs) and a C-terminal deaminase domain[3]. ADARs require duplex secondary structure in their substrate RNAs and use a base flipping mechanism to place the reactive adenosine into a zinc-containing active site[3,21]. Indeed, surface loops present on the ADAR deaminase domain have been identified that bind RNA on the 5′ side of an editing site (i.e. 5′ binding loop), bind RNA on the 3′ side of the editing site (i.e., 3′ binding loop), and is directly involved in base flipping (i.e., flipping loop)[21]. ADARs selectively edit certain adenosines over others in an RNA molecule and ADAR1 and ADAR2 have overlapping yet distinct selectivity[22–24]. However, our understanding of the basis for ADAR-specific selectivity is limited. Domain swapping experiments demonstrated that selectivity differences between the ADARs primarily originate from differences in their deaminase domains[25] and our earlier work indicated that differences in sequence in the 5′ binding loops of the deaminase domains were important determinants for ADAR-specific selectivity[26]. The structure and RNA recognition properties of the deaminase domain of human ADAR2 (hADAR2d) have been extensively studied[21,27,28]. Less is known about the ADAR1 deaminase domain (hADAR1d)[23,26,29]. This is, in part, due to the lack of structural information for the ADAR1 deaminase domain alone or in complex with RNA. Thus, studies that advance our understanding of structural features of the ADAR1 deaminase domain, particularly those that are unique to ADAR1, are important. Here we describe the use of high-throughput mutagenesis/functional screening along with biochemical studies of purified mutant proteins and Rosetta-based molecular modeling to define structural features unique to ADAR1. This includes the discovery and characterization of an unknown zinc-binding site present on the surface of the ADAR1 deaminase domain but absent in this domain from ADAR2. We identify amino acids that make up the ligand environment for this second zinc site and show these residues are important for efficient deamination by the ADAR1 deaminase domain in vitro and by full-length ADAR1 p110 in cultured human cells. Furthermore,

structural models constructed using Rosetta and constrained using results from our high-throughput mutagenesis and functional screening protocol explain previously observed properties of the ADAR1 deaminase domain and suggest roles for specific residues present in the ADAR1 5′ binding loop, including one mutated in AGS.

## Results

**Discovery of a second metal-binding site in hADAR1d.** Although the deaminase domains of human ADAR1 and ADAR2 (hADAR1d and hADAR2d, respectively) have 59% sequence similarity, these two proteins show different A-to-I editing efficiencies on different substrate RNAs[23,30]. There have been various studies directed at understanding the differences between these two proteins[23,24,26]. In addition, we have observed a substantial concentration-dependent aggregation of hADAR1d during purification that is not observed for hADAR2d under the same conditions. We reasoned that cysteine residues present in hADAR1d may be responsible for this phenomenon since there are a total of 11 cysteines in this domain (including the two putative catalytic zinc-binding cysteines at the active site), whereas there are a total of only six cysteines within hADAR2d. Cysteines are often important for catalytic activity or protein folding[31]. However, surface cysteines can also be involved in protein–protein aggregation and, in such cases, removing surface cysteines by mutagenesis can be helpful for experiments that require high protein concentrations[32–34]. Therefore, we chose to investigate the importance of cysteine residues present in hADAR1d. For this purpose, we used the Sat-FACS-Seq screening method that was developed by our lab[27]. This method utilizes Saturation mutagenesis (Sat) to incorporate codons for the 20 different amino acids at specific positions of interest, fluorescence-activated cell sorting (FACS) to sort yeast cells based on various levels of fluorescence from a fluorescent activity reporter, and next-generation Sequencing (Seq) to identify ADAR library mutants with different levels of editing activity. We generated a total of nine cysteine libraries for hADAR1d by saturation mutagenesis: C851, C893, C909, C976, C1081, C1082, C1129, C1169, and C1224, excluding the two putative catalytic metal-binding residues (C966 and C1036). Yeast cells were transformed with the hADAR1d cysteine codon plasmid libraries and fluorescent activity reporter plasmid. The fluorescent activity reporter contains a stop codon present in an ADAR1 substrate sequence upstream of yeast enhanced GFP coding sequence (Fig. 1a). Only when edited by hADAR1d mutants, can GFP be expressed, resulting in fluorescent yeast cells. Different fluorescence intensities observed correlate with A-to-I editing activity of the hADAR1d mutant present[27]. After inducing the yeast cells to overexpress hADAR1d mutants and reporters, we sorted cells into five different gates (R1–R5) based on different fluorescence levels. R1 contained cells with a background fluorescence level (expressing an inactive mutant of hADAR1d E912A) and R2–R5 gates contained cells having a fluorescence level above background (Fig. 1b). Cells in each gate were further cultured, followed by plasmid isolation to generate plasmid libraries from R1 to R5 gates.

Barcode PCR and sequencing with Illumina MiSeq were used to analyze the hADAR1d-encoding plasmids present in each gate. The resulting data were used to determine the enrichment levels of codons for different amino acids from R1 through R5 compared to the input library. This was accomplished by calculating an overall average fluorescence value ($F_{ave}$) for each amino acid at a specific position using the median fluorescence value of populations falling into each gate (R1 through R5) (See Supplementary Data 1). Then, each $F_{ave}$ was normalized to the wild type amino acid at that

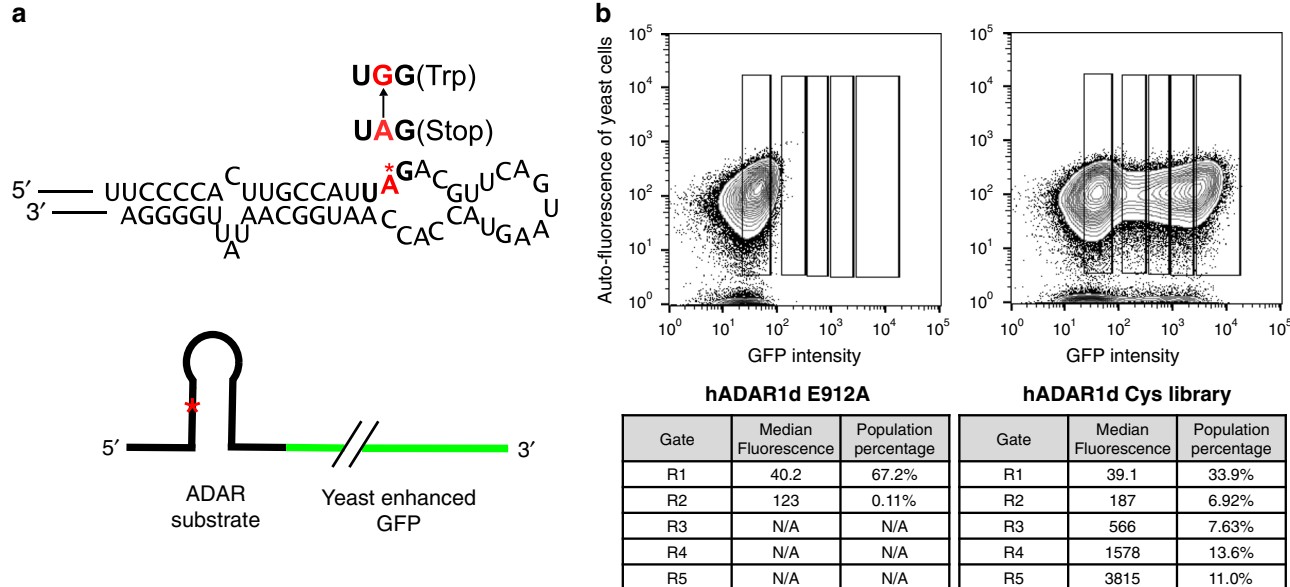

**Fig. 1 Fluorescence-activated cell sorting of cysteine libraries of hADAR1d. a** Schematic and ADAR substrate sequence of the fluorescent reporter used in Sat-FACS-Seq analysis. Editing site A is shown in red. **b** Fluorescence-activated cell sorting for hADAR1d E912A (deaminase inactive mutant) and hADAR1d cysteine libraries. hADAR1d E912A was used to determine the background level of fluorescence. Top: Cells expressing either yeGFP reporter and hADAR1d E912A or yeGFP reporter and hADAR1d Cysteine libraries were sorted into different gates based on the different levels of fluorescence. Bottom: Parameters corresponding to each sorting gate.

position to analyze the impact of all 20 amino acids at each position varied within hADAR1d (Fig. 2).

To determine which amino acids are preferred for high activity at each cysteine position, amino acids found in the R5 gate that have R5/input value over 1 were plotted as a logo plot (Fig. 3a). Most of the cysteine positions can tolerate mutations to many other amino acids other than the wild type residue, except for two positions, C1081 and C1082. The C1081 position showed a strong preference for the wild type cysteine residue. The C1082 position also has a high preference for cysteine, although it can be replaced with other amino acids and retain moderate activity. Thus, this Sat-FACS-Seq data indicated that C1081 and C1082 play an important role in the function of hADAR1d. Interestingly, we noted that the amino acids that are well tolerated at C1082X are histidine, glutamic acid, and aspartic acid (Fig. 3b). These amino acids are commonly found in metal-binding sites in proteins along with cysteine[35]. Because there are two adjacent cysteine residues at positions 1081 and 1082 and both positions showed a high preference for amino acids that are often associated with a metal binding, we speculated that an additional metal-binding site other than the catalytic zinc-binding site could exist in hADAR1d.

To test this hypothesis further, we generated a homology model of hADAR1d using the SWISS-MODEL web server[36] based on the crystal structure of hADAR2d bound to dsRNA[21]. The resulting homology model suggested that H1103 is in close proximity to C1081 and C1082, consistent with a metal-binding site involving ligation by the side chains of these three amino acids (Fig. 4a). Indeed, C1081, C1082, and H1103 are conserved among ADAR1 proteins from different organisms suggesting functional significance for all three (Fig. 4c, Supplementary Fig. 1). In addition, we used the Metal Ion-Binding Site Prediction and Docking Server (MIB)[37] to further assess the possibility of a second metal-binding site in hADAR1d. The homology model for hADAR1d was used as a template for the metal-binding residue prediction and docking for various metal ions with MIB. The C1081, C1082, and H1103 site showed a high binding score for

zinc, suggesting that these residues are involved in zinc binding. While the Sat-FACS-seq data were consistent with C1081 and C1082 being involved in metal binding, H1103 was predicted to be a part of the metal-binding site based on the homology model and the MIB prediction. To further test the role of H1103, we mutated this residue to various amino acids (H1103A, H1103F, H1103Q, H1103S, and H1103C) and determined the activity level for each mutant by monitoring fluorescence intensities for the fluorescent activity reporter that was used for Sat-FACS-Seq and normalizing to the fluorescence generated by hADAR1d WT (Fig. 4d). Interestingly, only H1103C, a mutant that could still bind metal ions, showed activity. Taken together these results provide strong support for a second zinc metal-binding site within the deaminase domain of hADAR1 involving C1081, C1082, and H1103.

In hADAR2d, Y561, Q562, and K578 residues correspond to hADAR1d C1081, C1082, and H1103 (Fig. 4b). In ADAR2, these residues stabilize the fold of the protein by a combination of H-bonding of the K578 ammonium group and the Q562 carboxamide along with hydrophobic interactions between the K578 methylenes and the Y561 phenyl ring (Fig. 4b). Therefore, it appears the second metal site in hADAR1d binding is responsible for the stabilization of the protein fold using metal binding instead of H-bonding and hydrophobic interactions as seen in the hADAR2d structure.

**Metal analysis of hADAR1d and hADAR2d by ICP-MS.** To measure metal content directly, both His$_{10}$-tagged hADAR1d WT and hADAR2d WT proteins were purified under the same conditions with 0.1 mM ZnCl$_2$ supplemented purification buffers. Ni-NTA purification, TEV protease cleavage to remove the His$_{10}$-tag followed by size exclusion chromatography were used to purify each protein sample. Excess zinc was removed at the final step of purification by using 0.5 mM EDTA and each protein was analyzed by ICP-MS to compare the metal content within each sample. For hADAR2d, we observed a 1:0.90 ratio of protein to zinc metal as expected given the known structure of this protein

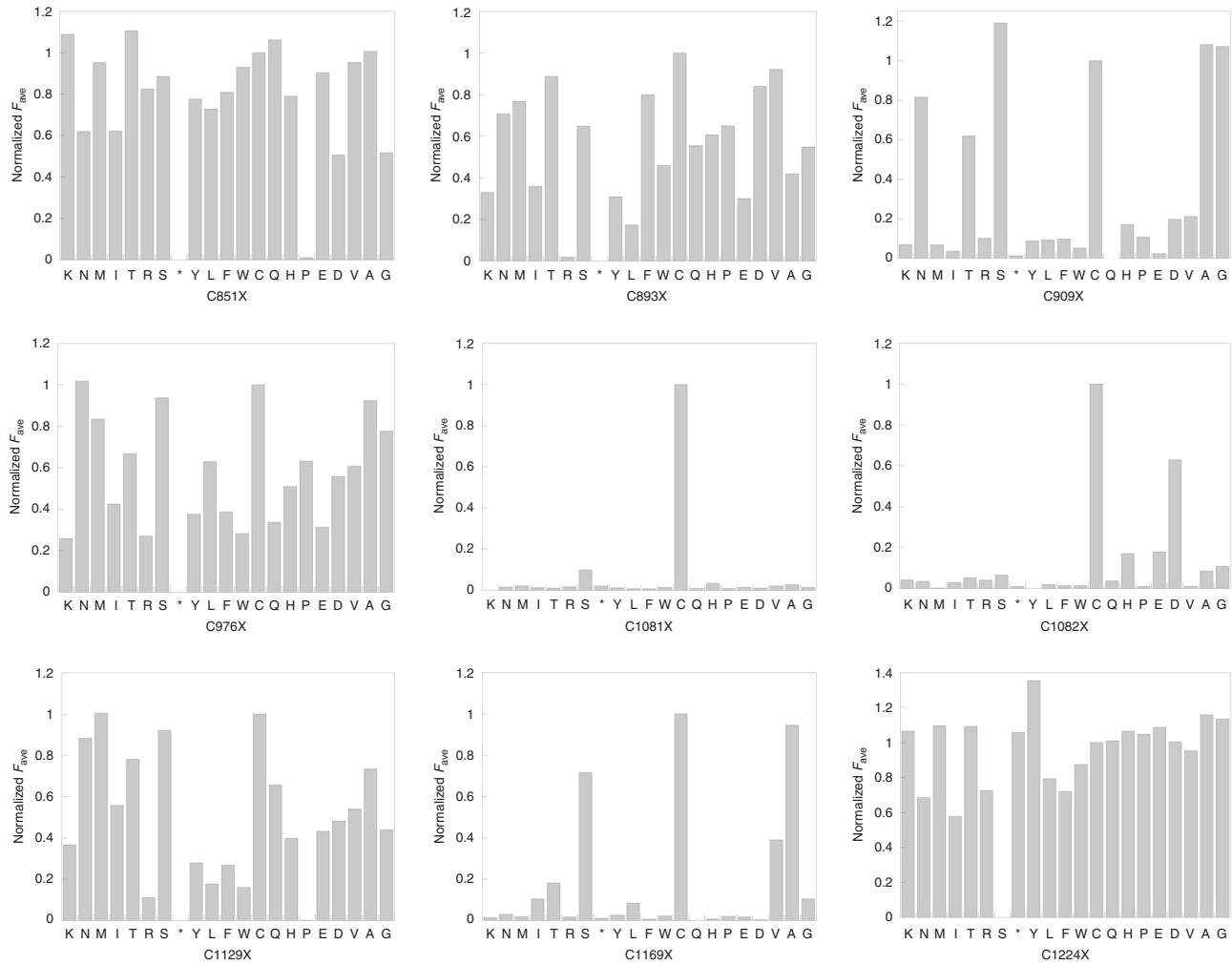

**Fig. 2 Normalized averaged fluorescence (Normalized $F_{ave}$) corresponding to each of 20 amino acids at nine different cysteine residues within hADAR1 catalytic domain.** Asterisk indicates stop codon.

containing one zinc ion at the active site[21] (Fig. 5a, Supplementary Table 1). For hADAR1d, a 1:1.93 ratio of protein to zinc metal was observed, consistent with two zinc-binding sites in this protein (Fig. 5a, Supplementary Table 1). In addition, native protein mass spectrometry analysis of hADAR1d WT revealed the protein mass corresponding to hADAR1d WT monomer, a cofactor (IP$_6$), and two zinc atoms (Fig. 5b, Supplementary Fig. 2). The observed mass is consistent with the calculated values within the errors of measurement for both intact (±1 amu) and native MS (±3 amu). This result along with the ICP-MS analysis strongly supports the presence of a second zinc-binding site within the ADAR1 catalytic domain.

For the ICP-MS studies described above, a His$_{10}$-tagged version of hADAR1d was used and protein concentration-dependent precipitation of this protein, and particularly for mutants of this protein, complicated its use. To facilitate additional biochemical studies with hADAR1d mutants, we generated a maltose-binding protein (MBP) tagged variant of hADAR1d. MBP is 42 kDa protein that is often used as a protein fusion tag because MBP can help solubilize proteins[38]. Both MBP-tagged hADAR1d wild type and metal-binding site mutants (hADAR1d C1082D, C1082E, and H1103D) were purified under the same conditions. The C1082 position was chosen for additional study since mutagenesis at this site to weak metal-binding amino acids, Asp and Glu, retained some enzymatic activity (Fig. 2).

MBP-hADAR1d WT and metal-binding site mutants were purified by amylose and heparin column using 0.1 mM ZnCl$_2$ supplemented purification buffers, then excess zinc was removed during a final step of purification by 0.5 mM EDTA. Each protein was then analyzed for metal content by ICP-MS. Interestingly, MBP-hADAR1d WT showed a ratio of 1:1.24 protein to zinc metal (Supplementary Table 1) suggesting this fusion protein is not completely bound to zinc at both sites. The lack of complete saturation of both ADAR1 zinc sites for the MBP fusion protein suggests the MBP domain is capable of solubilizing misfolded and demetallated ADAR1. Nevertheless, mutating the putative metal-binding residues, C1082 or H1103, to either Asp or Glu caused a substantial loss of zinc (Fig. 5c, Supplementary Table 1). In addition, mutating a different cysteine in the protein not predicted to be involved in the second zinc site (C893) to Asp minimally affected the measured zinc metal content compared to that of WT MBP-hADAR1d fusion (Fig. 5c, Supplementary Table 1). To gain additional insight into what metal is preferred at the second site, MBP-hADAR1d WT was purified without any metal ion supplementation in purification buffers. Only the amylose column was used to purify this protein sample to prevent demetallation caused by extensive purification steps. ICP-MS for this protein sample clearly showed only the presence of a substantial amount of zinc and no other metals (Fig. 5d, Supplementary Data 2).

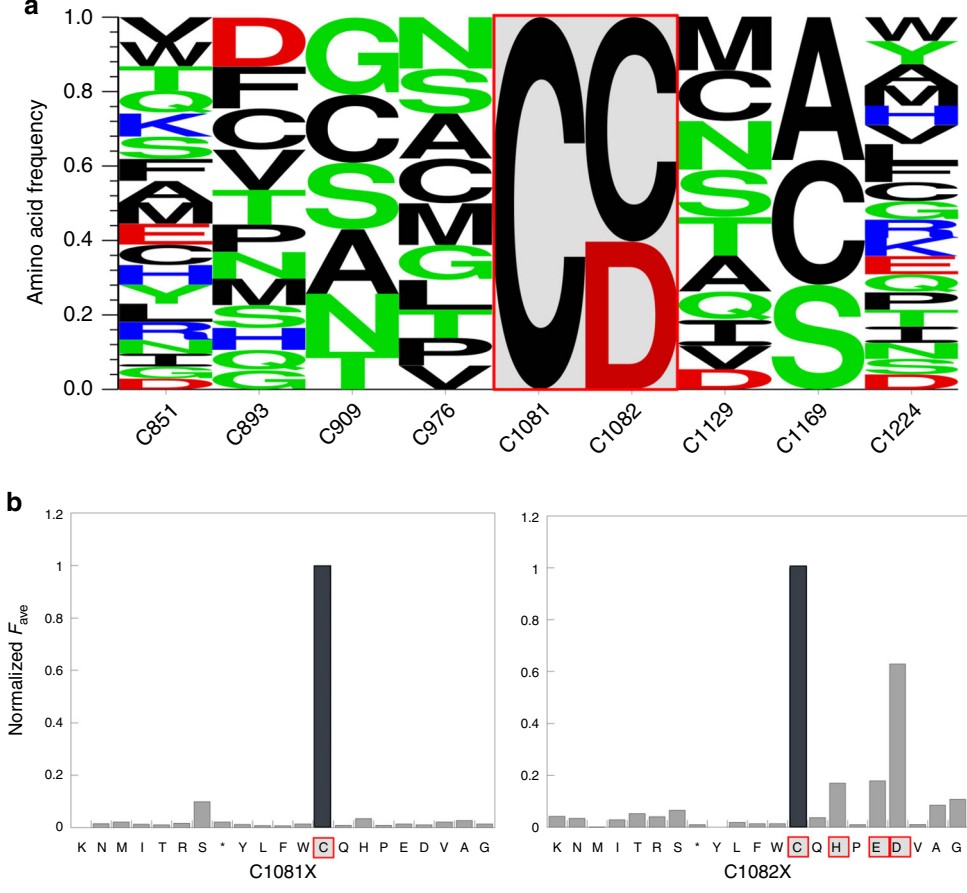

**Fig. 3 Sat-FACS-seq analysis of hADAR1d cysteine libraries. a** Logo plot result of Sat-FACS-Seq analysis of native Cys residues in hADAR1 catalytic domain. Positions where native Cys residue is highly preferred are boxed in red. **b** Normalized averaged fluorescence of position C1081 and C1082. Amino acids that are commonly involved in metal binding are boxed in red.

**Importance of the second metal-binding site for activity.** Editing activities of MBP-hADAR1d WT and the two metal-binding mutants (C1082D and C1082E) were compared using an in vitro deamination assay. We prepared a 147 nt RNA derived from the human Gli1 mRNA (hGli1 RNA) for this assay and we measured rates of conversion of A-to-I at a known editing site for each protein (Fig. 6a). MBP-hADAR1d WT showed $k_{obs} = 0.480 \pm 0.008 \, \mathrm{min}^{-1}$ and both metal-binding mutants, C1082D and C1082E, showed significantly lower catalytic activity compared to WT. The C1082D mutant was 5.7-fold slower with $k_{obs} = 0.084 \pm 0.004$, whereas C1082E mutant was 16-fold slower in activity having $k_{obs} = 0.03 \pm 0.01$ (Fig. 6b).

Next we tested the importance of the second zinc site for catalytic activity in the context of a full-length hADAR1 protein. For this purpose, full-length hADAR1 p110 WT and two metal-binding site mutants (C1082D and C1082E) were overexpressed in HEK293T cells. The expression level of each protein was confirmed by Western blot to ensure similar levels (Supplementary Fig. 3). Then, we measured editing levels for five different endogenous targets in HEK293T cells (Fig. 6c, Supplementary Fig. 4). For the endogenous Gli1 transcript, endogenous ADARs present in HEK293T cells edit this site to 24 ± 6%. When full-length hADAR1 p110 wide type was overexpressed, an increase to 51 ± 6% editing was observed, whereas the two metal-binding site mutants, C1082D and C1082E, showed lower editing (43 ± % and 24 ± 3%, respectively). For the other four endogenous targets (AZIN1 site 1 and 2, COG3, and NUP43), no editing was detected by endogenous ADARs. In addition, no editing was detected when the C1082D and C1082E mutants were overexpressed while

overexpression of WT clearly showed editing on those targets (AZIN1 site 1: 31 ± 2% and site 2: 61 ± 4%, COG3: 27 ± 2%, NUP43: 34 ± 2%). These results indicate that the second metal-binding site is important for the deaminase activity of full-length hADAR1 p110 in human cells.

**Homology modeling of hADAR1 catalytic domain.** A high-resolution structure of hADAR1 is not yet available. Nevertheless, our discovery here of a second zinc site and our previous high-throughput mutagenesis study of the hADAR1 5′ binding loop have provided useful constraints for modeling the hADAR1 deaminase domain structure[26]. Therefore, we generated a structural model of hADAR1d, using RosettaCM and the experimentally derived constraints described below[39–42].

The deaminase domains of hADAR1 and hADAR2 have high sequence similarity so structures of hADAR2d provide a good starting point for modeling hADAR1d. However, the sequence of the protein loop that interacts with the 5′ side of an RNA substrate (i.e., the 5′ binding loop) is substantially different in hADAR1 and hADAR2 (Fig. 7a). Due to the size of the loop, roughly 30 amino acids, lack of structural homologs, and no significant predicted secondary structural elements, additional data were desirable to reduce the structural search space and direct protein modeling efforts. In our previously published Sat-FACS-Seq studies of the hADAR1 and hADAR2 5′ binding loops[26,27], we noted a striking similarity in selectivity for specific amino acids at three common positions in the two hADARs (i.e., F457, D469, and R477 in hADAR2 and F972, D973, and K996 in hADAR1) suggesting these residues play similar roles in the two

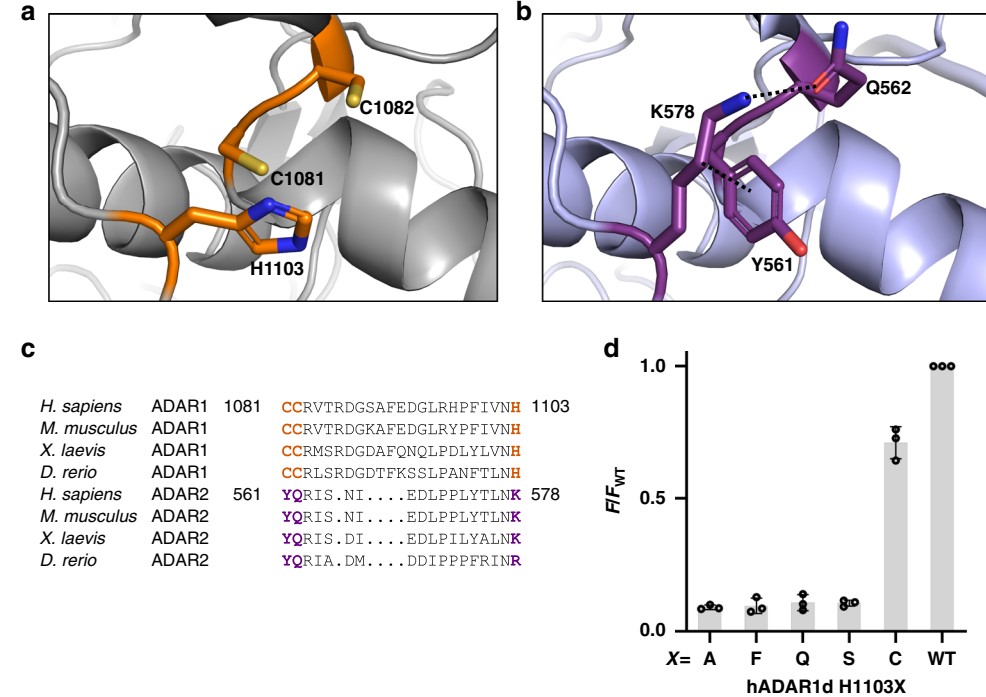

**Fig. 4 Prediction of metal-binding residues through a homology model and mutagensis study. a** Close-up view of possible second metal-binding site from a homology model of hADAR1d generated by SWISS-MODEL web server[36] where C1081, C1082, and H1103 are likely to be involved in a metal binding. **b** Crystal structure of hADAR2d (PDB: 5HP3)[21] with residues corresponding to possible metal-binding residues in hADAR1d. **c** Sequence alignment of ADAR1 and ADAR2 from different organisms showing C1081, C1082, and H1103 residues (highlighted in orange) and K561, Q562, and K578 (highlighted in purple) that are corresponding to possible metal-binding residues of hADAR1d WT. **d** Fluorescence-activated activity assay of hADAR1d H1103 mutants. BDF2-derived fluorescence reporter was used. F/F$_{WT}$ is the ratio of hADAR1d H1103 mutant fluorescence over hADAR1d WT fluorescence and data are plotted as the mean ± s.d. from three biological replicates. Individual data points are shown with black circles. Source data are provided in the Source Data file.

deaminase domains (Fig. 7b). In the hADAR2 5′ binding loop, D469 contacts R477 via an ion pair between the side chains and the phenyl ring of F457 provides a platform onto which the guanidinium group of R477 stacks in an apparent cation-π interaction. Thus, the side-chain positions of F972, D973, and K996 in hADAR1 were constrained in our Rosetta modeling to match the distances found between F457, D469, and R477 in structures of hADAR2 (Fig. 7c).

In addition to structurally conserved residues between hADAR1 and hADAR2, additional constraints were used in modeling to enforce canonical zinc coordination at the predicted binding site. Zinc ions are typically coordinated by four ligands forming tetrahedral structures[43,44]. However, the analysis described above only identified three hADAR1 residues involved in the second zinc-binding site (C1081, C1082, and H1103), leading to the question of the identity of a potential fourth metal-binding residue. Interestingly, our previous study of the 5′ binding loop of hADAR1 also provided evidence that a residue within that loop could serve this role[26]. We observed that mutation of H988 to cysteine within the 5′ binding loop of hADAR1d increased deaminase activity in our fluorescent reporter assay in yeast whereas mutation of this residue to each of the other common amino acids reduced activity[26] (Fig. 7b, Supplementary Fig. 5). Thus, H988 is a promising candidate for the fourth metal ligand given the fact that histidine and cysteine are common zinc-binding amino acids and the 5′ binding loop is near residues already implicated in the second zinc site[45]. To directly test this idea, MBP-hADAR1d H988D was prepared and analyzed by ICP-MS. As seen with the other metal-binding mutants (MBP-hADAR1d C1082D, C1082E, and H1103D), we observed the loss of metal upon the mutation of this residue,

consistent with this residue binding the second zinc (Fig. 5c). Therefore, Rosetta modeling was performed with the four hADAR1d residues C1081, C1082, H1103, H998 constraints in addition to the constraints derived from hypothesized structural homology to hADAR2d between residues F972, D973, and K996. For the residues constrained to the zinc ion, average distances from known structures containing a zinc ion bound by two cysteine and two histidine residues were used (Supplementary Table 2).

A three-dimensional model of the human ADAR1 deaminase domain was then generated using the RosettaCM protocol[39–42] including the constraints described above, all input and representative output structures are provided in the Supplementary information (Supplementary Fig. 6, 12, Supplementary Table 2). The crystal structure of hADAR2 deaminase domain bound to double-stranded RNA (PDB ID: 5HP3)[21] was used as the template for generating the model. During the modeling, zinc was treated as a ligand and incorporated as the homology models were constructed. Five-thousand models were generated from which the 10 lowest energy structures were evaluated (Supplementary Fig. 6 showing all 10 overlaid). Figure 8a shows a representative hADAR1d model superimposed on our previously reported crystal structure of an hADAR2d-RNA complex[21]. Among the top 10 lowest energy models, this model shows the most similar conformation of the 5′ binding loop interactions observed in hADAR2d crystal structure that were used as modeling constraints (Fig. 8c, Supplementary Fig. 7). Also, in this model, the long hADAR1 5′ binding loop folds back onto the deaminase domain to engage in zinc binding coordinated by C1081, C1082, H1103, and H988. In fact, this conformation of the 5′ binding loop observed in hADAR1d homology model is

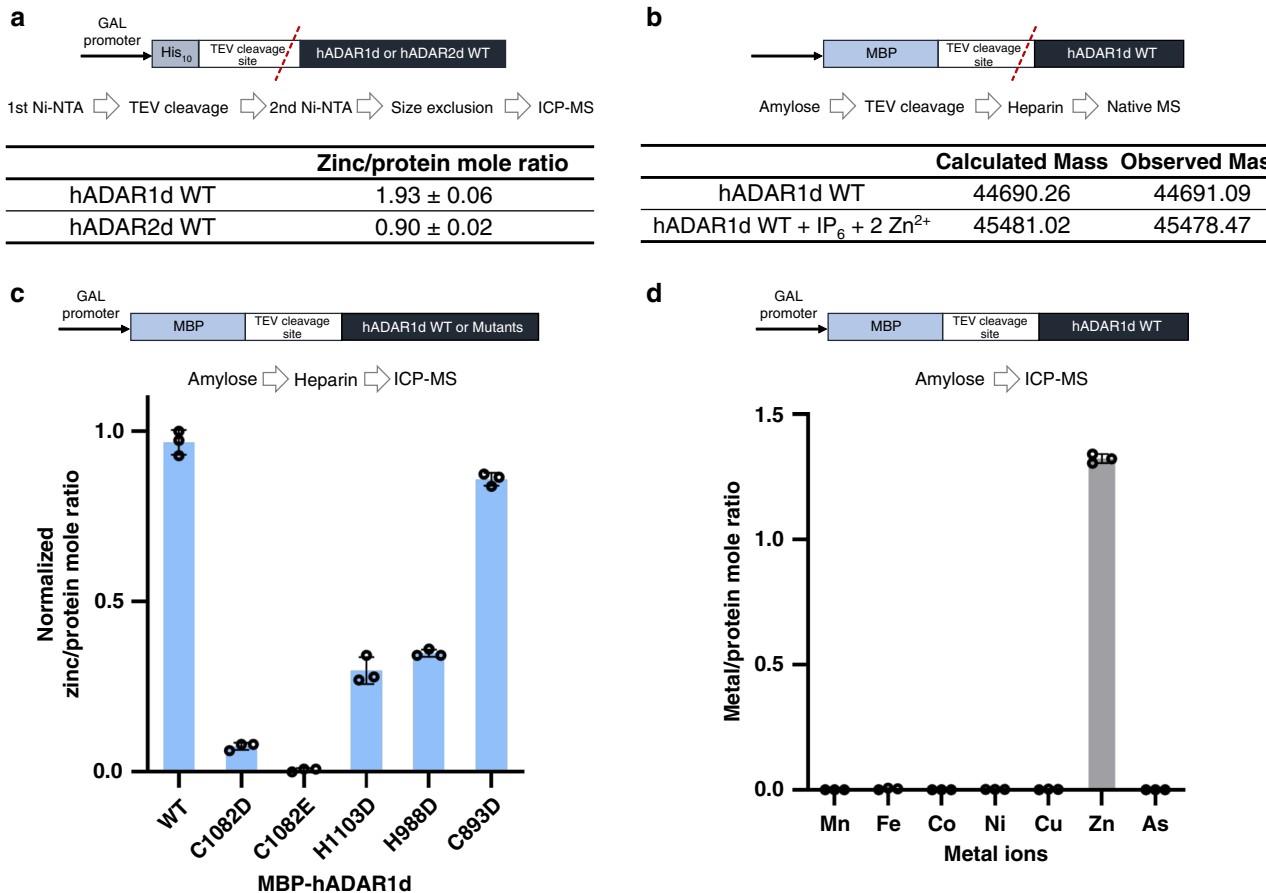

**Fig. 5 Metal analysis results of hADAR1d by ICP-MS and protein native MS analysis.** Protein construct design and purification steps are shown for each experiment. **a** Zinc/protein mole ratio of $His_{10}$ -tag fusion of hADAR1d WT and hADAR2d WT as control represented in mean ± s.d. from three independent experiments. **b** Results of protein native MS. Observed mass and calculated mass of intact LC-MS (top row) and native LC-MS (bottom row) analysis are shown in the table. The observed mass is consistent with the calculated values within the errors of measurement and ambiguity in the protonation state of $IP_6$ that is bound to the protein. **c** Zinc/protein mole ratio of MBP fusion hADAR1d WT and mutants (C1082D, C1082E, H1103D, H988D, and C893D). **d** Metal/Protein mole ratio from metal screening ICP-MS result of MBP-hADAR1d WT that was purified without metal supplementation. Data in (**c**), and (**d**) are plotted as the mean ± s.d. from three independent experiments. Individual data points are shown with black circles.

substantially different from what is seen in hADAR2d crystal structure, which is located close to the RNA (Fig. 8a–c). To test this structural model further, we turned to chemical cross-linking mass spectrometry (XL-MS) using the MS-cleavable cross-linker disuccinimidyl dibutyric urea (DSBU)[46,47]. Several cross-linked peptides were detected in the XL-MS analysis of DSBU-treated hADAR1d (See Supplementary Data 3 for raw cross-linking spectrum files). Indeed, four specific cross-linked peptides were reproducibly identified in three different experimental replicates (Supplementary Fig. 8, 9, and Supplementary Table 3). These results are in good agreement with our homology model, showing a Cα–Cα distance of the two cross-linked residues between 10 and 25 Å, which is the distance typically observed from cross-linking mediated by DSBU[46,47]. Importantly, cross-linking between S986 and K1105 supports the conformation of the 5′ binding loop represented in hADAR1d homology model since S986 is located in the 5′ binding loop only two amino acids away from a metal-binding residue, H988 (Fig. 8d, e).

The structural model presented here is also consistent with the known properties of the protein. For instance, given the positioning of the hADAR1 5′ binding loop in the model, most of the amino acids present on this loop are directed away from the likely RNA-binding site, unlike the hADAR2 5′ binding loop where the side chains of H471 and R474 contact substrate RNA on the edited strand ~10 nt 5′ to the editing site (Figs. 7c, 8b).

The hADAR1 deaminase domain readily deaminates RNA substrates with short 5′ duplexes (<10 bp) whereas a longer 5′ duplex (>10 bp) is required for the efficient editing of the hADAR2 deaminase domain[26]. In addition, the 5′ binding residues that are not evolutionarily conserved among ADAR1 proteins from different species (corresponding to aa978-aa987 in human, Fig. 7a, Supplementary Fig. 1) are located away from the region predicted to be the RNA-binding surface in our model. This hADAR1 deaminase domain model also suggests likely roles for specific residues in the 5′ binding loop. For instance, K974 within the 5′ binding loop is a good candidate for an RNA-binding residue given its proximity to the predicted RNA-binding surface (Fig. 8b, Supplementary Fig. 10a). Also, our model predicts methylenes present on the side chain of K999, a residue mutated in Aicardi-Goutierres Syndrome (K999N), is involved in hydrophobic contacts to Y1208, resulting in stabilization of the 5′ binding loop fold (Supplementary Fig. 10c).

Although the model described here provides valuable information on the general course of the ADAR1 5′ binding loop and features of the second metal-binding site, it should be noted that the ten lowest energy models do not converge to a single conformation throughout the loop (Supplementary Fig. 6). The uncertainty for the exact positions for certain residues in this loop highlights the need for additional studies to define high-resolution structures for ADAR1 bound to RNA.

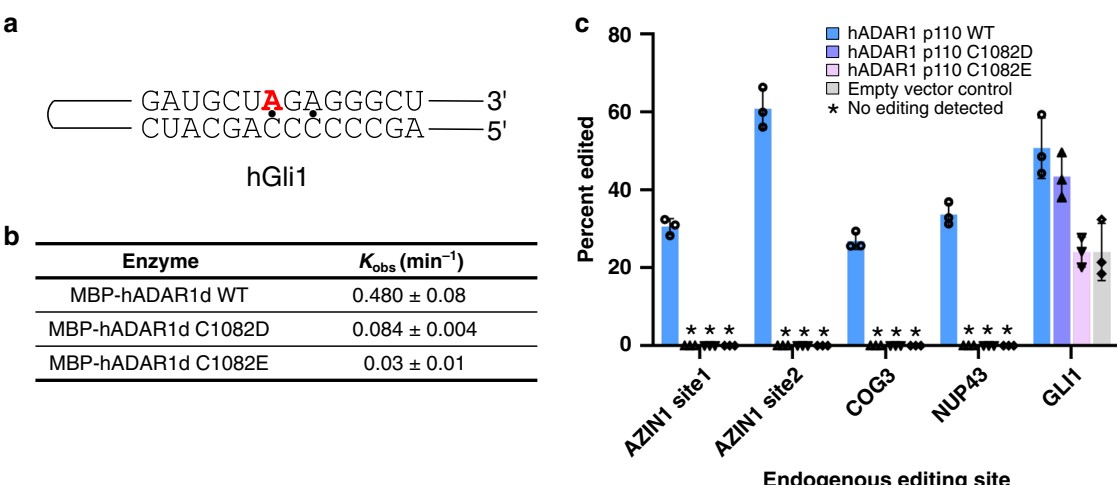

**Fig. 6 Analysis of catalytic activity of hADAR1 showing importance of the second metal-binding site. a** Sequence of hGli1 RNA substrate used for in vitro deamination kinetics study. Highlighted in red is an editing site. **b** MBP-hADAR1d WT and metal-binding mutants (C1082D and C1082E) deamination kinetics. hGli1 RNA (10 nM) and 750 nM of each protein were used. Data are presented as the mean ± s.d. from three independent replicates. **c** Endogenous target site editing of hADAR1 p110 WT and metal mutants (C1082D and C1082E) in HEK293T cells. Individual data points from hADAR1d p110 WT, C1082D, C1082E, and empty vector control are represented as circles, up-pointing triangles, down-pointing triangles, and diamonds, respectively. Asterisk indicates no editing observed. Data are plotted as the mean ± s.d. from three biological replicates. Source data for Figs. 6b, c are provided in the Source Data file.

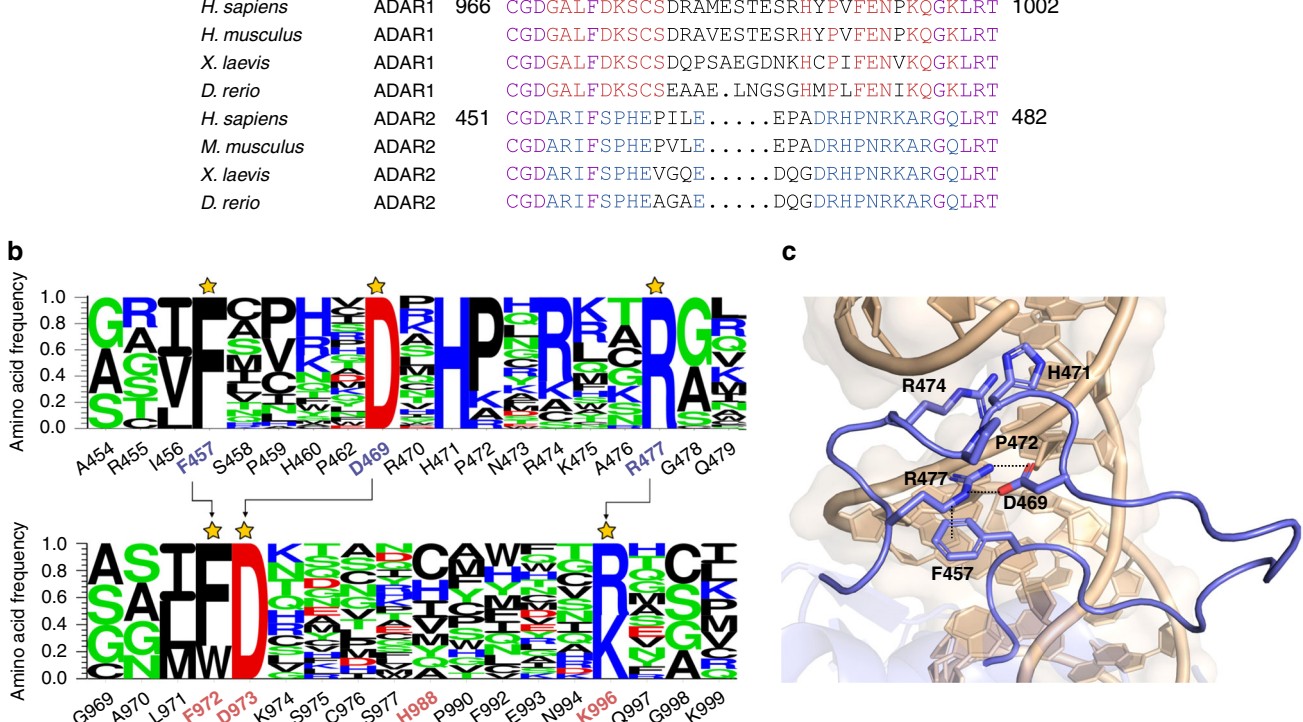

**Fig. 7 Comparison of the 5′ binding loops of hADAR1 and hADAR2. a** Sequence alignment of the 5′ binding loop of ADAR1 and ADAR2 from different organisms. Purple: conserved in both ADAR1 and ADAR2, Red: conserved in ADAR1, Blue: conserved in ADAR2, Black: not conserved. **b** Comparison of Sat-FACS-Seq results for the 5′ binding loops of hADAR1[26] and hADAR2[27]. Arrows indicate functionally similar residues in the different loops. **c** Close-up view of interactions within the 5′ binding loop observed in the crystal structure of hADAR2d with dsRNA[21].

## Discussion

Metal ions are important cofactors that can serve essential roles in protein structure and function[48,49]. In this work, we discovered a previously undisclosed zinc-binding site within catalytic domain of human ADAR1 that, along with the zinc ion present in the active site, makes for a total of two known zinc sites in this

domain. We also identified likely coordinating amino acids for the second zinc site (H988, C1081, C1082, and H1103) and demonstrated that mutation of these residues results in reduced zinc content for the protein. Mutation of these residues also reduces RNA editing activity. Given the positioning of the second zinc site, we believe it most likely that this site plays a structural

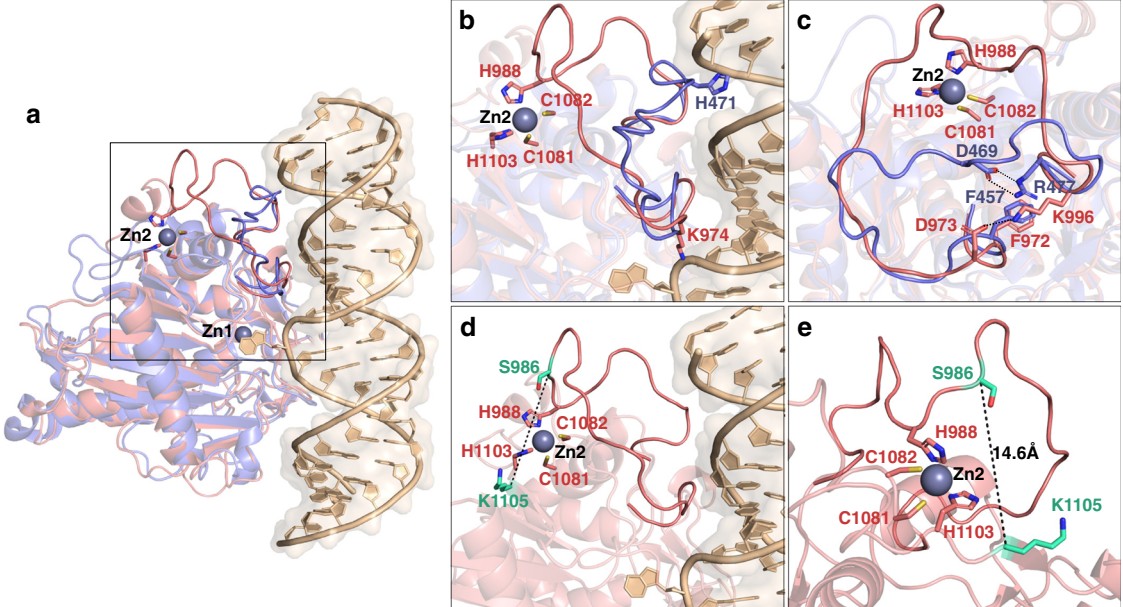

**Fig. 8 Comparison of a hADAR1d model with the crystal structure of an hADAR2d-RNA complex[21] and chemical cross-linking mass spectrometry (XL-MS) using a DSBU cross-linker. a** A representative hADAR1d model (in salmon) is superimposed on crystal structure of a hADAR2d-RNA complex (in blue and RNA in wheat), showing the 5′ binding loop of each protein and the second metal-binding site of hADAR1d. Zn1 is a catalytic zinc at the active site and Zn2 is bound in the second metal site reported here. **b** Close-up of the 5′ binding loop of each protein and the second metal-binding site of hADAR1d. H471 residue in hADAR2d is a RNA contacting residue seen in the crystal structure of hADAR2d. K974 is a possible RNA contacting residue predicted from the hADAR1d model structure. **c** Close-up comparison of the 5′ binding loop conformations of hADAR1 and hADAR2 and ionic and cation-π interactions of residues within each of the 5′ binding loop. **d** DSBU cross-linking between K1105 and S986 supports the conformation of the 5′ binding loop observed from hADAR1d model structure. **e** Close-up view of the cross-linking of K1105-S986 nearby the second zinc-binding site, showing Cα-Cα distance of 14.6 Å.

role like other zinc sites found on the surface of proteins[50,51]. Indeed, our model suggests the second zinc site is important for maintaining the conformation of the 5′ binding loop suitable for RNA substrate recognition (Fig. 8a–c). Importantly, our characterization of hADAR1 mutants defective in zinc binding at the second site indicates this site is important for editing activity (Fig. 6). These results suggest this location in the protein could be targeted for small molecule inhibitor development, perhaps by incorporating zinc-binding groups into candidate ligands[52–54].

Beyond the apparent structural role, the second zinc site in hADAR1 could be involved in additional functions. For instance, it is possible that when each ligating amino acid (H988, C1081, C1082, and H1103) is bound to the second zinc site, the fully active conformation of the 5′ binding loop is populated. However, if H988 binding is transient and it releases from the metal site, the 5′ binding loop could adopt lower activity conformations. In addition, it is well known that cysteines in proteins are susceptible to oxidation under oxidative stress conditions[55]. Cysteine oxidation can lead to protein conformational changes, unfolding, and degradation[55–59]. Finally, the hADAR1 second zinc site could serve as an interface for protein–protein interactions[60]. Additional studies are required to determine if the zinc site identified here has other regulatory roles or is involved in the formation of protein complexes.

To gain additional understanding of the possible structural impact of this previously unknown zinc site in hADAR1, we carried out molecular modeling using Rosetta and structures of the hADAR2 deaminase domain bound to RNA[21] along with constraints defined from high-throughput mutagenesis/functional screening (i.e., Sat-FACS-Seq)[26,27]. It is important to note the assumptions we made in defining our modeling constraints. First, based on the remarkable similarity in amino acid preference at three positions in the two different hADARs 5′ binding loops (e.g., F457, D469 and R477 in ADAR2; F972, D973, and K996 in

ADAR1) (Fig. 7b, c), we assumed these amino acids play similar roles in stabilizing the 5′ binding loop conformation in each protein. Second, we assumed that the side chain of H988 is directly involved in binding to the second zinc in hADAR1. This is based on the Sat-FACS-Seq result at this position (Fig. 7b, Supplementary Fig. 5) previously reported by us[26] indicating that mutation to cysteine enhanced hADAR1d activity whereas mutation to other residues reduced activity. In fact, this assumption was confirmed later by the metal analysis, indicating that H988 is also involved in a metal binding (Fig. 5c). Rosetta modeling is often supplemented with experimentally derived constraints[61]. For example, chemical cross-linking mass spectrometry (XL-MS) has been used to generate constraints for modeling[62] as well as other structural related experimental data such as NOE measurements from NMR[63] and EPR/DEER measurements[64,65]. Our modeling approach is unique in that our constraints were obtained from comparative high-throughput mutagenesis/functional screening instead of structural data alone. The model we arrived at not only supports the biochemical data reported previously[26], but also allows for analysis of possible roles for different 5′ binding loop residues. Understanding the interactions of the 5′ binding loops present in the ADAR proteins is crucial to understanding ADAR selectivity[25,26]. However, it has been challenging to predict the RNA-binding residues present in the hADAR1 5′ binding loop given only hADAR2-RNA structural data. Indeed, our model suggests the 5′ binding loop structures diverge from each other substantially and this conformational difference between hADAR1d and ADAR2d was further supported by the XL-MS analysis (Fig. 8). Part of the 5′ binding loop of hADAR2 is in position for direct contact to the RNA substrate ~10 nt from the editing site, whereas a similar contact with hADAR1 seems unlikely given that a large portion of its 5′ binding loop is directed away from the RNA (Fig. 8a, b).

However, the model does predict K974 is a likely RNA contact residue given its proximity to the putative RNA-binding site (Fig. 8b). While additional studies will be necessary to confirm this prediction, the fact that hADAR1 editing activity is maintained when this site is mutated to polar residues (e.g., R, N, Q, T, and H) is consistent with K974 functioning in direct RNA binding[26] (Fig. 7b, Supplementary Fig. 10a, b). On the other hand, our model predicts the side-chain methylenes of K999 contact Y1208, stabilizing the 5′ binding loop fold (Supplementary Fig. 10c). This is consistent with our previously published mutagenesis data indicating that K999 can be mutated to several different large hydrophobic residues (e.g., I, L, M, and V) and editing activity is maintained (Fig. 7b, Supplementary Fig. 10c, d)[26]. This also suggests that the AGS-associated mutation (K999N) disrupts the protein structure by introducing a short polar side chain into this hydrophobic site.

The unique structural features of ADAR1 described here increase our knowledge of this protein in general and provide the basis for understanding differences between hADAR1 and hADAR2. Furthermore, the unique zinc site in hADAR1 suggests the possibility of undisclosed regulatory mechanisms and innovative ways to control its activity.

## Methods

**Preparation of Cys library and FACS.** hADAR1d cysteine library in hADAR1d E1008Q (in YEpTOP2P-GAL1 vector) was prepared by saturation mutagenesis using QuikChange II XL site-directed mutagenesis kit (Agilent) and the resulting mutagenesis products were transformed into XL-10 Gold ultracompetent E. coli cells (Agilent) following the manufacturer's protocol (Primers used for saturation mutagenesis are listed in Supplementary Table 4). To ensure coverage of the scope of each library, E. coli colonies at least 10 times greater number than the library diversity (NNS = 32 possible codons) were pooled (at least 300 E. coli colonies). Plasmid samples from each saturation mutagenesis libraries were pooled in an equal amount to make the start plasmid library[27]. S. cerevisiae INVSc1 strain (Invitrogen) was sequentially transformed with the yeGFP reporter plasmid containing BDF2-derived sequence with TRP1 selection followed by library plasmids (hADAR1d Cys libraries) with URA3 selection. For ADAR library plasmids transformation, yeast cells with a yeGFP reporter plasmid were used for three parallel transformations using a highly efficient lithium transformation protocol[66]. For each transformation, $1.0 \times 10^8$ cells and 3 μg of the start plasmid library were used. After transformation, about 1/200 volume of cells from each transformation was plated on CM – ura – trp + 2% glucose plate to determine the transformation efficiency and the remaining cells from three parallel transformations were combined and used to inoculate 10 ml of CM – ura – trp + 2% glucose media for overnight growth at 30 °C. After 20 h of induction, cells were collected by centrifugation for cell sorting analysis[27]. Catalytically inactive mutant (hADAR1d E912A) was also used as a negative control and processed the same way. Cells with hADAR1d Cys library and hADAR1d E912A were diluted in PBS to 20,000 cells/μl and sorted using Beckman Coulter Astrios EQ cell sorter at UC Davis flow cytometry shared resource laboratory. GFP excitation was at 488 nm and emission at 529/28 nm. Cells expressing hADAR1d Cys library were collected in a BD falcon polypropylene round-bottom tubes corresponding to each gate (R1–R5) based on the fluorescence level. More than 150,000 cells (in around 2 ml) were collected for each gate and each collected cell was used to inoculate equal volume of CM – ura - trp + 2% glucose media for 24 h growth, after which the whole culture was used to inoculate 13 ml of CM – ura - trp + 2% glucose media for another 24 h growth. Cells were then pelleted and plasmids were isolated from each cell pellet. Contour plots creation and data analysis were performed using FlowJo.

**Library preparation and Illumina sequencing.** hADAR1d sequence containing each Cys residue (159–201 nt) was amplified from the plasmid libraries isolated from yeast cells by two-step PCR using Phusion hot-start DNA polymerase (Thermo Fisher Scientific) to prepare Illumina Miseq sequencing samples (Primers are listed in Supplementary Table 5). The first PCR used primers containing overhang adapter sequence and amplicon-specific sequence. The PCR products were purified by agarose gel and QIAquick Gel Extraction Kit (Qiagen) and used for a second PCR with primers containing a leader sequence (P5, P7), an 8 bp barcode specific to each sample, and a sequence overlapping with the adapter sequence in the primers used in the previous PCR. Final PCR products were purified by agarose gel and QIAquick Gel Extraction kit (Qiagen), pooled in an equal amount, and sequenced by Illumina Miseq PE250.

**NGS sequencing data processing.** Paired-end reads from Miseq were demultiplexed according to the sample-specific barcodes. Using Trimmomatic,

adapter sequences were removed and reads were trimmed on both sides with a quality cutoff of 5, which then further filtered with an overall average quality cutoff of 30 and checked by FastQC. The resulting forward reads were long enough to cover target codon sequences, thus only the forward reads were utilized for further data analysis. For each position of variation, the abundance of each codon was determined by searching 12 nt sequences including target codon, the positional barcode (silent mutation), and six extra nucleotides in the 5′ direction using MobaXterm command line. To eliminate bias caused by the difference in the total number of reads among different samples, the total number of reads in each sample (input library, R1–R5) were adjusted with the smallest reads number. Then, the abundance of amino acids in R1 through R5 was weighed against their abundance in the input library to determine enrichment levels in each gate. Frequencies of amino acids enriched in R5 gate were calculated from their enrichment levels and plotted using Seq2 Logo 2.0 server[67]. Average fluorescence $F_{ave}$ for each amino acid at each position was estimated using the equation below[27] (Supplementary Data 1) using Microsoft Excel and was used to generate bar graphs using KaleidaGraph.

$$F_{ave} = \frac{\sum_{i=1}^{5} (\text{enrichment level in R}i \times \text{mean fluorescence of R}i)}{\sum_{i=1}^{5} (\text{enrichment level in R}i)}$$

**Expression and purification of His₁₀-tagged hADAR1d WT and hADAR2d WT.** Protein expression and purification of His₁₀-tagged hADAR1d WT and hADAR2d WT were carried out following a previous protocol[21,68] with modifications. Each protein plasmid (in YEpTOP2P-GAL1) was transformed into S. cerevisiae BCY123 cells and plated onto a selection plate (CM –ura + 2% glucose). A single colony was used to inoculate a 5 ml CM – ura + 2% glucose media for overnight growth at 30 °C. The resulting culture of 10 ml was used to inoculate a 1 L CM – ura -trap + 3% glycerol + 2% lactate media and grew until OD₆₀₀ of the culture reaches 1–2 (~20 h). Cells were induced by adding galactose to a final concentration of 3% for 6 h, harvested by centrifugation, and stored at – 80 °C. Cells were lysed in lysis buffer (20 mM Tris-HCl, pH 8.0, 5% glycerol, 750 mM NaCl, 30 mM imidazole, 0.1 mM ZnCl₂, 1 mM BME, 0.05% Triton X-100, cOmplete EDTA-free protease inhibitor (Roche)) using a microfluidizer followed by centrifugation (43,667 × g for 50 min) to obtain clear cell lysate. Cell lysate was loaded to 5 ml Ni-NTA agarose Superflow Cartridges (Qiagen) and washed with 50 ml wash I buffer (20 mM Tris-HCl, pH 8.0, 5% glycerol, 750 mM NaCl, 30 mM imidazole, 0.1 mM ZnCl₂, 1 mM BME), then 100 ml of wash II buffer (20 mM Tris-HCl, pH 8.0, 5% glycerol, 350 mM NaCl, 30 mM imidazole, 0.1 mM ZnCl₂, 1 mM BME). Protein was eluted by a linear gradient of 30–400 mM imidazole in wash II buffer and pooled for His₁₀-tag cleavage using 1-to-2 ratio of TEV protease to the protein by initial 1 h incubation at room temperature, followed by overnight dialysis in a buffer containing 20 mM Tris-HCl, pH 8.0, 5% glycerol, 350 mM NaCl, 50 mM imidazole, 0.1 mM ZnCl₂, 1 mM BME at 4 °C using a 10 K MWCO dialysis cassette (Thermo Fisher Scientific). The protein mixture was passed through another 5 ml Ni-NTA column and the flow-through and 100 ml wash (20 mM Tris-HCl, pH 8.0, 5% glycerol, 350 mM NaCl, 50 mM imidazole, 0.1 mM ZnCl₂, 1 mM BME) were collected, concentrated to 1 ml, and dialyzed against 20 mM Tris-HCl, pH 8.0, 5% glycerol, 350 mM NaCl, 70 mM imidazole, 0.1 mM ZnCl₂, 1 mM BME overnight at 4 °C. The concentrated protein was further purified by a gel filtration on a GE Healthcare HiLoad 16/600 Superdex 200 pg column using a superdex buffer (20 mM Tris-HCl, pH 8.0, 5% glycerol, 350 mM NaCl, 70 mM imidazole, 0.1 mM ZnCl₂, 1 mM BME) and dialyzed against 50 mM Tris-HCl, pH 8.0, 10% glycerol, 200 mM KCl, 70 mM imidazole, 0.5 mM EDTA, 0.01% NP-40 substitute, 1 mM DTT overnight at 4 °C, which then concentrated to 22 μM for hADAR1d WT and 122 μM for hADAR2d WT for subsequent ICP-MS metal analysis.

**Expression and purification of MBP-tagged hADAR1d WT and mutants.** A codon-optimized gene fragment of MBP for optimal expression in S. cerevisiae was ordered from GeneArt (Thermo Fisher Scientific). Cloning of MBP-tagged hADAR1d WT was carried out using Gibson master mix (New England Biolabs) following the manufacturer's protocol. The resulting MBP-tagged hADAR1d WT construct was used for mutagenesis to generate metal-binding mutants (MBP-hADAR1d C1082D, C1082E, H1103D, and H988D) as well as control mutant (MBP-hADAR1d C893D) using QuikChange II XL site-directed mutagenesis kit (Agilent) following manufacturer's protocol. (See Supplementary Fig. 11 for gene sequence of MBP-hADAR1d WT and Supplementary Table 6 for primers used for cloning). Protein was expressed in S. cerevisiae BCY123 cells as described above. Lysate obtained as described above was passed through 3 ml Amylose column (New England Biolabs) and washed with 50 ml of each wash buffers: wash I buffer (20 mM Tris-HCl, pH 8.0, 5% glycerol, 1 M NaCl, 0.1 mM ZnCl₂, 1 mM BME), wash II buffer (20 mM Tris-HCl, pH 7.5, 5% glycerol, 500 mM NaCl, 0.1 mM ZnCl₂, 1 mM BME), wash III buffer (20 mM Bis-Tris, pH 7.0, 5% glycerol, 75 mM NaCl, 0.1 mM ZnCl₂, 1 mM BME). The target protein was eluted with a 10 mM maltose in wash III buffer (20 mM Bis-Tris, pH 7.0, 5% glycerol, 75 mM NaCl, 0.1 mM ZnCl₂, 1 mM BME) and further purified on a 5 ml HiTrap Heparin HP column (GE Healthcare Lifesciences) by washing with 50 ml of heparin wash buffer (20 mM Bis-Tris, pH 7.0, 5% glycerol, 75 mM NaCl, 0.1 mM ZnCl₂, 1 mM BME) and eluting with a linear gradient of 75 mM – 1 M NaCl in heparin wash buffer. The purified protein was pooled and dialyzed against 50 mM Tris-HCl, pH 8.0, 10% glycerol, 200 mM KCl, 0.5 mM EDTA, 0.01% NP-40 substitute, 1 mM DTT overnight at 4 °C, which then concentrated to 9–232 μM for ICP-MS analysis and

deamination kinetics. MBP-tagged hADAR1d WT protein used for metal screening analysis was purified by amylose column and dialyzed as described above using the identical buffers without ZnCl$_2$ supplementation, which was then concentrated to 129 μM for ICP-MS metal analysis.

**Inductively coupled plasma-mass spectrometry (ICP-MS)**. Each protein sample (20–125 μl volume with different concentrations) and the equal volume of final storage buffer of each sample as a buffer blank were submitted in three replicates to the Interdisciplinary Center for Plasma-Mass Spectrometry at the University of California at Davis for sample digestion and analysis using an Agilent 8900 ICP-MS (Agilent Technologies) primarily for the determination of Zn concentrations in a minimal volume digest. Buffer blanks for each sample replicate aliquot was digested in parallel and analyzed along with the samples. One set of digestion batch quality control (QC) samples were prepared at the same volumes as the replicate sample aliquots for each sample type and consisted of a Method Blank (MB = 18.2 MΩ/cm MilliQ Water), Laboratory Control Sample-A (LCS-A = NIST 1640a Trace Elements in Natural Water (National Institute of Standards and Technology), and Laboratory Control Sample-B (LCS-B = 1 ppm ICV custom calibration standard mix (Inorganic Ventures, Inc.). The digestion batch QC samples were digested in parallel and analyzed along with the samples and buffer blanks. The digestion procedure was performed as follows: MB, LCS-A, LCS-B, protein samples, and blank buffers were digested with 20 or 40 μL conc. tr. metals grade HNO$_3$ (Fisher Scientific) with heating at 95 °C on a Fisher Scientific ISOTEMP 125D hot block for 0.5 h. Then, 75 μL 30% ULTREX II Ultrapure Reagent H$_2$O$_2$ (J.T. Baker) was added incrementally (25, 25, and 25 μL over a 10 min period) to hot samples at 95 °C, followed by heating at 95 °C for 15 min after the final addition. Samples were allowed to cool, then brought to FV = 250 or 500 μL with MilliQ water resulting a 8% HNO$_3$ matrix.

Digested samples, digestion batch QC, external standards, or blanks were mixed at ~18:1 ratio with a custom internal standard solution using a mixing tee, then introduced in the Agilent 8900 ICP-MS via peristaltic pump at 0.10 rps using a 0.4 mL/min MicroMist nebulizer to produce an aerosol in a 2 °C temperature-controlled double pass spray chamber leading to a 1550 W plasma. Samples, buffer blanks, digestion batch QC, external standards, and blanks were injected using an Agilent SPS 4 Autosampler equipped with a 0.25 mm ID sample probe, except for the 250 μL FV samples, buffer blanks, and digestion batch QC, which were injected manually. The custom internal standard solution with Sc, Ge, Y, In, and Bi was diluted from Inorganic Ventures single element standards to 7.5 ppm of each element. External standards were diluted from the ICV custom calibration standard mix to 0.1, 0.5, 1, 5, 10, 50, 100, 500, and 1000 ppb for Ti, Cr, Cu, Zn, and Ba calibrations, while Mg, Si, and P were at concentrations 10× higher in each calibration level. The ICP-MS instrument was tuned and calibrated prior to analysis and operated in MS/MS mode using a 3-point peak pattern with three replicates per injection and 50 sweeps per replicate. Since the analysis method was optimized for Zn and minimal volume injections, the integration time for Zn was 0.5 s and all of the measured masses had 0.1 s integration time. He mode was used in the collision/reaction cell during the measurements. NIST 1640a Trace Elements in Natural Water, a single element Inorganic Ventures Zn standard diluted to 50 ppb, and a blank were analyzed initially for independent source QC calibration and blank verification. The ICV custom calibration standard mix was analyzed at 1, 10, and 100 ppb (10× higher concentrations for Mg, Si, and P) every 9th sample or less as quality controls along with a blank to monitor instrument performance and provide continuing calibration and blank verification. All standards were prepared in 8% HNO$_3$ (v:v, conc. tr. metals grade HNO$_3$: 18.2 MΩ/cm MilliQ Water). External standards and samples were prepared in 15 mL polypropylene centrifuge tubes and the internal standard solution were prepared in 50 mL polypropylene centrifuge tubes (Fisherbrand). The raw data was processed using Agilent MassHunter ICP-MS software and further analyzed to calculate Metal/protein mol ratio using Microsoft Excel (Supplementary Table 1 and Supplementary Data 2).

**Intact and native LC-MS assay**. For protein used in native MS assay, hADAR1d WT with TEV cleavable, N-terminal MBP tag was inserted into pFastBac1 vector (Invitrogen), transformed and expressed in High Five cells (Thermo Fisher). Protein was purified similar to *S. cerevisiae* MBP-hADAR1d WT with an additional step introduced to cleave MBP tag using TEV protease prior to the Heparin column. The protein sequence after TEV cleavage of MBP-hADAR1d used for native MS is SGGS (L833-V1226). Buffers were modified to exclude 0.01% Triton X-100, 1 mM BME, and replaced with 1 mM TCEP to reduce signal suppression and artefacts. Purified protein at 10 μM stored in 25 mM HEPES, 150 mM NaCl, 1 mM TCEP, 0.5 mM EDTA, 5% Glycerol, pH 8.0 was used for intact and native MS analysis. The sample was centrifuged at 16,200 × g for 10 min at 20 °C prior to LC-MS analysis to remove any particulate aggregates. All LC-MS measurements were performed on an Agilent 1290 Infinity II UPLC-6530 Accurate-Mass Q-TOF instrument in positive ion mode using electrospray ionization. For mass determination under intact LC-MS conditions, UPLC separation was performed using ZORBAX 300SB-C8 column (50 mm × 2.1 mm, 5 μm) (Agilent). The mobile phase consisted of solvent A (0.2% formic acid in water) and solvent B (0.2% formic acid in acetonitrile). The sample was eluted with the following gradient at a flow rate of 0.3 mL/min:95% A:5% B (0–1 min), 5% A:95% B (1–3 min), hold until 4 min, and return to 95%A:5% B by 0.1 min. The column and auto sampler temperature were

40 °C and 6 °C, respectively. The sample flow from the column was directly coupled to Q-TOF and mass measured in electrospray ionization mode (nozzle 2000V, fragmentor 150 V, sheath gas 350 °C, drying gas 325 °C). For native LC-MS, UPLC separation using a PolyHYDROXYETHYL ATM column (200 × 2.1 mm, 5 μm) (PolyLC) was conducted with an isocratic elution of 200 mM ammonium acetate (pH 7.0) at a flow rate of 0.1 mL/min. The column was held at 23 °C and the auto sampler at 6 °C. Acquisitions were performed in the *m/z* range 500–3200 for intact MS and in the *m/z* range 500–8000 for the native MS experiments. External TOF mass calibration was obtained over the *m/z* range 50–3,200 using a solution of 1 μM purine, 25 μM ammonium trifluoroacetate, 0.25 μM HP-0921 in 95%:5% acetonitrile:water mixture prior to the analysis. Data analysis was performed using Agilent MassHunter software.

**Deamination kinetics for MBP-hADAR1d**. Deamination kinetics were carried out using hGli1 RNA prepared by in vitro transcription (see Supplementary Method) under the single turnover condition with 15 mM Tris-HCl, pH 7.4, 20 mM KCl, 40 mM potassium glutamate, 5 mM NaCl, 4% glycerol, 1.5 mM EDTA, 0.003% Nonidet P-40, 0.5 mM DTT, 1.0 μg/ml yeast tRNA, 160 U/mL RNAsin, 10 nM hGli1 RNA, and 750 nM of each MBP-hADAR1d. Each reaction was incubated at 30 °C for 30 min before adding protein. Reactions were quenched at each given time point by adding 190 μl of hot water followed by vortexing and incubation at 95 °C for 5 min. cDNA was obtained via RT–PCR using Access RT-PCR core kit (Promega) (primers used for RT-PCR are listed in Supplementary Table 7) and sequenced by Sanger sequencing, which then quantified with Chromas software to obtain the editing level. The $k_{obs}$ (min$^{-1}$) of each assay was calculated using the equation $[P]_t = \alpha[1 - e^{(-k_{obs} \cdot t)}]$, where $[P]_t$ is the percentage of edited product at time $t$, $\alpha$ is the fitted end point, and $k_{obs}$ is the observed rate constant. Each experiment was carried out in three replicates.

**Endogenous editing in HEK293T cells and analysis**. HA tagged full-length hADAR1 p110 WT in pcDNA 3.1 vector was used to prepare mutants (hADAR1 p110 C1082D and C1082E) by site-directed mutagenesis using QuikChange II XL site-directed mutagenesis kit (Agilent) following the manufacturer's protocol (Primers used for mutagenesis can be found in Supplementary Table 6). HEK293T cells were cultured in Dulbecco's modified Eagle's medium, 10% fetal bovine serum, and 1% anti-anti at 37 °C, 5% CO$_2$. Once cells reached 70–90% confluency, $6.4 \times 10^3$ cells were seeded into 96-well plates. After 24 h, cells were transfected with 750 ng of hADAR1 plasmids using Lipofectamine 2000 (Thermo Fisher Scientific). More specifically, each plasmid and Lipofectamine 2000 were mixed with Opti-MEM Reduced Serum Media (Thermo Fisher Scientific), which subsequently combined and incubated for 20 min before the solution was added to each designated well. Cells after transfection were incubated at 37 °C, 5% CO$_2$ for 48 h. Total RNA was extracted and collected using RNAqueous Total RNA Isolation Kit (Thermo Fisher Scientific) and RQ1 RNase-free DNase (Promega) was added for DNA digestion. Total RNA samples were used for Nested RT-PCR using Access RT-PCR kit (Promega) for 20 cycles followed by Phusion Hot Start DNA Polymerase (Thermo Fisher Scientific) for the second PCR of 30 cycle (Primers used for nested RT-PCR in Supplementary Table 8). cDNA products were purified by agarose gel and QIAquick Gel Extraction kit (Qiagen), which then submitted for Sanger Sequencing. The editing level was quantified with Chromas software and a bar graph was generated by Graphpad Prism 8. The experiments were performed in biological triplicate.

**Homology modeling of hADAR1d WT**. A three-dimensional model of hADAR1d catalytic domain was generated in three stages using the RosettaCM protocol[39]. The crystal structure of hADAR2 deaminase domain bound to dsRNA (PDB:5HP3)[21] was used as the template for generating the model. Promals3D was used to generate a sequence alignment of the template and query sequence to correlate sequence position to structure[41]. First, the query sequence was threaded onto the template sequence, resulting in a threaded partial model. To fill in unaligned regions, Monte Carlo sampling was used to generate Rosetta de novo fragments[40]. The scoring function of this sampling was a combination of the Rosetta low-resolution energy function and distance constraints from the template structure. Second, Monte Carlo sampling was again used for full backbone minimization. During the third stage, the structure underwent full-atom refinement for side-chain optimization and refinement of the side-chain and backbone conformations. During the third stage, side-chain constraints were used to enhance sampling[42]. The conserved residues in the 5′ binding loop identified through Sat-FACS-Seq[26] were constrained to mimic an ionic interaction (D973 and K996) and cation-pi stacking interaction (K996 and F972) observed in the hADAR2d structure[21]. Furthermore, zinc was treated as a ligand and H988 was defined as the fourth metal-binding residue when the structural models were constructed (Supplementary Figs. 6 and 7). The metal-binding constraints were defined using average distances and angles from 20 different PDBs found from MetalPDB[69,70] each containing zinc bound by two Cys and two His residues (Supplementary Fig. 12, Supplementary Table 2). Using this protocol, 5000 decoys of hADAR1d structure were generated and the 10 lowest energy structures were evaluated. All input files and the lowest 10 energy structures are available in GitHub.

**Chemical cross-linking with DSBU and in-gel tryptic digestion.** His[10]-tagged hADAR1d WT was purified by Ni-NTA column as described above. Chemical cross-linking with DSBU was carried out according to previous protocols[46,47] except for a few modifications. The protein sample was buffer exchanged with cross-linking buffer (20 mM HEPES, pH 8.0, 350 mM NaCl, 5% glycerol, 1 mM TCEP) using Zeba spin desalting column (Thermo Fisher Scientific) to remove incompatible buffer components. A final concentration of 10 μM of the buffer exchanged protein and 10 mM of DSBU (1000× molar excess, Thermo Fisher Scientific) in a final volume of 10 μl solution was incubated at room temperature for 1 h. Three independent cross-linking reactions were carried out in parallel. Each reaction was quenched by adding 2 μl of 1 M Tris-HCl, pH 8.0, and incubated at room temperature for 30 min. A total of 12 μl reaction solution was analyzed by SDS-PAGE gel (Invitrogen), which then stained with Imperial Protein Stain (ThermoFisher Scientific) following the manufacturer's protocol. A protein gel band that corresponds to M.W. of the protein (44 kDa) was cut out for the in-gel digestion, which was carried out according to the previous procedure[71]. In brief, the gel pieces were washed with 200 μL each of $H_2O$ and acetonitrile (ACN), followed by drying the gel pieces in miVac (SP Scientific) for 5 min. Samples were further incubated with 100 μL of 10 mM DTT for 50 min at 55 °C and washed with 200 μL of ACN. Free cystine was alkylated with 100 μL of 50 mM IAA for 20 min at room temperature in the dark, which then washed with 200 μL each of 100 mM ammonium bicarbonate and ACN, alternatively. After two cycles of washing, 40 μL of a 0.025 μg/μL trypsin solution were added to the dried samples and incubated for 30 min to allow the gels to expand, followed by addition of 200 μL of 100 mM ammonium bicarbonate buffer. The tryptic digestion was performed at 37 °C for 18 h, and the samples were dried in miVac.

**LC-MS analysis and Data analysis for chemical cross-linking.** The digested samples were reconstituted in $H_2O$ and characterized using the UltiMate™ WPS-3000RS nanoLC system coupled with Orbitrap Fusion Lumos (Thermo Fisher Scientific). Each sample (1 μL) was injected, and the analytes were separated on Acclaim™ PepMap™ 100 C18 LC Column (3 μm, 0.075 mm × 250 mm, Thermo Fisher Scientific) at a flow rate of 300 nL/min. Water containing 0.1% formic acid and 80% acetonitrile, and containing 0.1% formic acid was used as solvents A and B, respectively, with the LC gradient: 0–90 min, 4–35% (B); 90–110 min, 35–45% (B); 110–115 min, 45–100% (B); 125–126 min, 100–4% (B). MS spectra were collected with a mass range of $m/z$ 300–1800 in positive ionization mode. The filtered top ten precursor ions were subjected to fragmentation through 30 ± 3% higher-energy C-trap dissociation (HCD) with nitrogen gas, and the same precursor within 45 s was excluded. The precursor and product ions were detected using orbitrap at 120,000 resolution ($R = 120,000$) and 15,000 resolution ($R = 15,000$), respectively. The raw spectra were converted to.mgf file using pXtract and searched against MeroX[47]. C-terminals of lysine and arginine were used for specific cleavage sites, and missed cleavages were restricted to 3. Precursor and fragmentation mass tolerance were limited to 20 ppm, the carbamidomethylation at cysteine was assigned as the fixed modification, and the oxidation at methionine was selected as the variable modification. The default settings of DSBU cross-linker and RISEUP Mode were used to identify the cross-linked peptides with high confidence through filtering 5% false discovery rate (FDR) and the overall score <50.

**Reporting summary.** Further information on research design is available in the Nature Research Reporting Summary linked to this article.

## Data availability

Illumina sequencing data are available in the NCBI Sequence Read Archive (SRA) under accession code PRJNA590991. The cross-linking mass spectrometry (XL-MS) data, and intact and native mass spectrometry data have been deposited to the Proteome Xchange Consortium via the MassIVE partner repository with the data set identifier PXD021052 [https://doi.org/10.25345/C5T758] and PXD021175 [https://doi.org/10.25345/C5VV0W], respectively. All input files and the lowest 10 energy structures of ADAR1d are available in GitHub at https://github.com/siegel-lab-ucd/Publication_Tiffy/tree/master/High-throughput%20Mutagenesis%20Reveals%20Unique%20Structural%20Features%20of%20Human%20ADAR1. All data is available from the corresponding author upon reasonable request. Source data are provided with this paper.

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

## Acknowledgements

The authors acknowledge funding from the US NIH, grant R01GM061115 (to P.A.B). This project was supported by the University of California Davis Flow Cytometry Shared Resource Laboratory with funding from the NCI P30 CA093373 (Cancer Center), and NIH NCRR C06-RR12088, S10 OD018223, S10 RR12964 and S10 RR 026825 grants and with technical assistance from Ms. Bridget McLaughlin and Mr. Jonathan Van Dyke. Research conducted by J.B.S. and Y.Z. was supported by the University of California-Davis, the National Science Foundation Award Numbers 1827246, 1805510, 1627539, the National Institute of Environmental Health Sciences of the National Institutes of Health (NIH) under Award Number P42ES004699, NIH Award Number R01 GM 076324-11, and the Rosetta Commons. E.E.D. was supported by training grant T32-GM113770 from the National Institutes of Health. The content is solely the responsibility of the authors and does not necessarily represent the official views of the National Institutes of Health or National Science Foundation.

## Author contributions

E.E.D., Y.Z., and J.B.S. generated a homology model of hADAR1d using Rosetta. Y.X. and C. B.L. analyzed chemical cross-link samples by tryptic digestion, LC-MS/MS, and data analysis. A.K.P. and F.F. conducted protein intact and native MS and data analysis. A.K. constructed YEpTOP2P-GAL1 vector containing MBP-tagged hADAR1d WT and designed the initial purification strategy for MBP-hADAR1d WT. S.P. conducted all the other experiments not listed above and wrote the initial manuscript draft. S.P. and P.A.B. edited the manuscript.

## Competing interests

A.K.P. is an employee of and has ownership interest in Agios Pharmaceuticals. P.A.B. is a consultant for Agios Pharmaceuticals, ProQR Therapeutics and Beam Therapeutics. P.A.B. has ownership interest in Beam Therapeutics.
