## [Peer Review File · Nature Communications]

Reviewers' comments:

Reviewer #1 (Remarks to the Author):

Park et al present a high throughput mutagenesis and modelling study applied to ADAR1, an enzyme that deaminates adenosine to inosine in RNA and which has recently been identified as an important target for anti-cancer drug development.

The study uses an innovative screening method to do saturation mutagenesis at specific sites in the protein. While the initial motivation seems to be to find cysteine residues that could be mutated to improve the behaviour of the protein during purification, the authors stumbled on a potential Zinc ion binding site. This was then followed up with by ICP-MS which confirmed that ADAR1, unlike ADAR2, has a second bound zinc ion.

This led the authors to undertake further modelling of the ADAR1 catalytic domain, using the zinc binding site and other constraints based on their mutagenesis data and evolutionary constraints. Overall the model provides some insights into differences in substrate specificities between ADAR1 and ADAR2.

The experimental work presented here is well executed and the authors provide elegant reasoning for the constraints that they apply to their modelling. While this work is insightful, I'm not convinced that the advances it provides are sufficient to merit publication in Nature Communications.

Some specific points are below:

1. The identification of the second Zn²⁺ site is supported by the modelling, which suggests an additional two histidine residues that might contribute to zinc ion binding. The authors should confirm this observation by showing that mutations at H1103 and H988 that lose activity also lose zinc binding using ICP-MS.
2. While the modelling looks very compelling, it would benefit from additional data orthogonal to the mutagenesis data and the modelling calculations. An alternative methodology like cross-linking mass spectrometry could add additional constraints, particularly if different types of cross-linkers (BS3, EDC) were used. Alternatively, if the authors were able to collect SAXS data for this domain and show that the best models match SAXS curves better than alternatives, this would give more weight to the current model.
3. The authors suggest that K974 might be important for RNA binding, however, their own mutagenesis data suggest that the identity of this residue is of limited importance so long as it is polar. This suggests that their model has limited capacity to identify residues that are likely to interact with RNA.
4. Parts of the discussion are highly speculative. For example, the authors suggest that the newly identified zinc ion binding site might promote protein-protein interactions, similar to those observed in

APOBECs. It is not clear from this statement if the site identified in this work is in a conserved position with respect to APOBECs. Furthermore, if this part of the surface were required for protein-protein interactions, one would expect that the surface around this site should show higher conservation than other parts of the surface of ADAR1 catalytic domain.

Reviewer #2 (Remarks to the Author):

Although 3D structural information on the ADAR2 catalytic domain has been available, we have very limited information on the ADAR1 structure. In this study, the authors conducted high-throughput mutagenesis of the ADAR1 deaminase domain (hADAR1d) and identified two cysteines critical for A-to-I editing activity. Furthermore, they proposed that these two cysteines are involved in chelation of a Zinc ion and stabilization of the protein structure. They determined the number of Zn ions bound to ADAR1d by ICP-MS and claimed two Zn ions binding. It has been previously known that hADAR1d contains a Zinc ion at the catalytic center. Thus, their findings indicate that ADAR1d contains two Zinc ions, one for catalytic process and another for protein structure stabilization. The authors carried out Rosetta-based protein structure modeling based on the known ADAR2 catalytic domain structure, and revealed an RNA contacting loop structure (5' binding loop fold), involving C1081, C1082, and H1103, which all chelate the newly found Zn ion. The study also revealed a possible interaction between K999 and Y1208, stabilizing the 5' binding loop fold. The importance of the loop structure is further emphasized by the presence of Aicardi-Goutières Syndrome (AGS)-associated mutation K999N, which perhaps disrupts this critical loop fold. These new findings on the ADAR1 catalytic domain structure are very informative and are likely to appeal to general readers of Nature Communication. However, there are several issues, which must be addressed.

Major Concerns:

The authors did not provide sufficient explanation for their interpretation of ICP-MS results (Figure 5).

- 1) They obtained Zinc/protein mole ratios of 2.67 and 1.07 for ADAR1d and ADAR2d, respectively, and concluded that ADAR1d contains two Zn ions (Figure 5a).
- 2) The ICP-MS result 2.67 seems to be more reasonably rounded up to 3.0 instead of down to 2.0, and I do not understand the reasoning behind the authors decision to report two, instead of three, Zn ions bound to ADAR1d.
- 3) In fact, mutagenic analysis of 9 cysteines indicated that not only C1081 and C1082 but also C909 and C1169 are functionally important (Figure 2), perhaps indicating that they are involved in binding to a third Zn ion (?).

Because of difficulty of solubilizing ADAR1d mutant proteins, they prepared all mutant proteins as

maltose-binding protein fusion proteins (MBP-ADAR1d) to be used for ICP-MS analysis and in vitro editing assay.

1) They obtained Zinc/protein mole ratio of 1.24 for MBP-ADAR1d-WT, indicating that only one Zn ion (presumably catalytic center Zn) is retained in this fusion protein. Yet, MBP-ADAR1d-WT exhibited full editing activity on Gli1 RNA in vitro ((Figure 6b). The result indicates that the second (or even third) Zn ion is not essential.

2) Furthermore, Zinc/protein mole ratios of 0.251 and 0.176 were obtained for MBP-ADAR1d-C1082E and ADAR1d-C1082E, respectively. One interpretation of these results is that both Zn ions (catalytic center and second one) are lost in these mutants. Perhaps the loop structure consisting of the second Zn and C1081, C1082, and H1103 somehow affects the stability of the catalytic center structure and retention of the catalytic center Zn ion. Loss of the catalytic center Zn ion for these two mutants could easily explain their total loss of editing activities in vitro (Figure 6b) and in vivo (Figure 6c).

The authors should employ an alternate strategy, other than direct ICP-MS to confirm the exact number of Zn ions bound to ADAR1d. They should acquire the Zn-binding data needed for a classical Scatchard plot, which would provide not only the stoichiometry of Zn binding, but also affinity constants and any significant interaction among the 2 (or 3) binding sites. The binding data for this plot could be obtained using ICP-MS to observe the competition between bound "natural" zinc and one of its stable isotopes when the protein is titrated with a solution of the latter. Or perhaps isothermal titration calorimetry might be used, though this involves the risk of denaturing the protein when it is stripped of zinc. In any case some additional support is needed for the authors' claimed stoichiometry.

Reviewer #3 (Remarks to the Author):

Review of Park et al.,

In this manuscript the catalytic domain of ADAR1 is investigated. The authors use a mutagenesis approach to interrogate residues located in the 5' binding region of the catalytic domain to predict a zinc-binding region.

Zinc-binding is further verified by ICP-MS and mutants in the second zinc coordinating site are being tested for activity in a collection of editing substrates.

Modeling is used to predict the structure of the 5' interacting region and to overlay the catalytic domains of ADAR1 and ADAR2.

The manuscript is well written and the flow of experiments presented appears logic, convincing, and complete.

I only have a few comments that should be addressed:

*) Why did the authors only focus on C1082 when testing for their impact on catalytic activity (Fig 6)? Were other mutants tested as well?

*) an extended alignment of the ADAR1 catalytic domain would help to get a better idea of the structural elements involved (alpha helices, beta sheets, 5' Interacting region, catalytic core). Such an alignment could be provided in the supplements and would allow the reader to get better oriented.

*) would it be possible to replace the 5' interacting region of ADAR1 with that of ADAR2? (I realize that this may be far-fetched but if a functional hybrid enzyme can be generated it would allow many predictions to be tested).

Regards,

Michael Jantsch

Below is a point-by-point response to the reviewers' comments. The entire reviewer report is reproduced below in italics. Our responses follow each specific point raised.

Reviewer #1 (Remarks to the Author):

Park et al present a high throughput mutagenesis and modelling study applied to ADAR1, an enzyme that deaminates adenosine to inosine in RNA and which has recently been identified as an important target for anti-cancer drug development.

The study uses an innovative screening method to do saturation mutagenesis at specific sites in the protein. While the initial motivation seems to be to find cysteine residues that could be mutated to improve the behaviour of the protein during purification, the authors stumbled on a potential Zinc ion binding site. This was then followed up with by ICP-MS which confirmed that ADAR1, unlike ADAR2, has a second bound zinc ion.

This led the authors to undertake further modelling of the ADAR1 catalytic domain, using the zinc binding site and other constraints based on their mutagenesis data and evolutionary constraints. Overall the model provides some insights into differences in substrate specificities between ADAR1 and ADAR2.

The experimental work presented here is well executed and the authors provide elegant reasoning for the constraints that they apply to their modelling. While this work is insightful, I'm not convinced that the advances it provides are sufficient to merit publication in Nature Communications.

Our response: The position that the advances are not sufficient to merit publication in Nature Communications was not shared by the other two reviewers. Furthermore, we believe the additional results provided in the revised manuscript serve to strengthen our conclusions and increase the overall impact of the work.

Some specific points are below:

1. The identification of the second Zn²⁺ site is supported by the modelling, which suggests an additional two histidine residues that might contribute to zinc ion binding. The authors should confirm this observation by showing that mutations at H1103 and H988 that lose activity also lose zinc binding using ICP-MS.

Our response: As suggested, an additional ICP-MS experiment was carried out using two histidine mutants (H1103D and H988D) and we observed loss of metal caused by these mutations (Fig. 5c). Although we did not measure the activity of these two mutants using *in vitro* deamination kinetics, our fluorescence activity assay of H1103X (Fig. 4d) shows that only histidine to cysteine mutation at 1103 position retains catalytic activity whereas mutation with other amino acids (A, Q, F, and S) results in loss of activity. Additionally, we observed a decrease in activity with the H988D mutation in our Sat-FACS-seq analysis that involved saturation mutagenesis at this site (Supplementary Fig. 4).

2. While the modelling looks very compelling, it would benefit from additional data orthogonal to the mutagenesis data and the modelling calculations. An alternative methodology like cross-linking mass spectrometry could add additional constraints, particularly if different types of cross-linkers (BS3, EDC) were used. Alternatively, if the authors were able to collect SAXS data for this domain and show that the best models match SAXS curves better than alternatives, this would give more weight to the current model.

Our response: We agreed that an additional orthogonal experiment like crosslink/mass spectrometry (CL-MS) would further support our model. Therefore, this was carried out using an MS cleavable crosslinker, DSBU. From this experiment, we identified four pairs of cross linked amino acids reproducibly identified in all experimental replicates. These are in good agreement with our model (Fig 9c, d and Supplementary Fig. 7).

3. The authors suggest that K974 might be important for RNA binding, however, their own mutagenesis data suggest that the identity of this residue is of limited importance so long as it is polar. This suggests that their model has limited capacity to identify residues that are likely to interact with RNA.

Our response: We believe our model is already providing useful information regarding the role of different conserved amino acids in the ADAR1 deaminase domain. As described in the discussion section of our manuscript, our earlier mutagenesis study of two conserved lysine residues within the 5' binding loop (K974, and K999) showed a puzzling and seemingly conflicting result: K974 prefers a polar residue whereas K999 prefers a hydrophobic residue. It was not possible to rationalize this difference for two highly conserved lysines without a high quality model of the protein structure. However, the model reported here indicates that K974 is a potential RNA contacting residue since it is proximal to the RNA. Each of the polar mutants accommodated at this position could bind RNA by H-bonding. In contrast, the CH₂ groups in the K999 side chain are predicted to make a hydrophobic contact with Y1208 (Supplementary Fig. 8). Only hydrophobic mutants would be expected to function similarly.

4. Parts of the discussion are highly speculative. For example, the authors suggest that the newly identified zinc ion binding site might promote protein-protein interactions, similar to those observed in APOBECs. It is not clear from this statement if the site identified in this work is in a conserved position with respect to APOBECs. Furthermore, if this part of the surface were required for protein-protein interactions, one would expect that the surface around this site should show higher conservation than other parts of the surface of ADAR1 catalytic domain.

Our response: We have removed much of this speculative discussion, including the reference to APOBECs.

Reviewer #2 (Remarks to the Author):

Although 3D structural information on the ADAR2 catalytic domain has been available, we have very limited information on the ADAR1 structure. In this study, the authors conducted high-throughput mutagenesis of the ADAR1 deaminase domain (hADAR1d) and identified two cysteines critical for A-to-I editing activity. Furthermore, they proposed that these two cysteines are involved in chelation of a Zinc ion and stabilization of the protein structure. They determined the number of Zn ions bound to ADAR1d by ICP-MS and claimed two Zn ions binding. It has been previously known that hADAR1d contains a Zinc ion at the catalytic center. Thus, their findings indicate that ADAR1d contains two Zinc ions, one for catalytic process and another for protein structure stabilization. The authors carried out Rosetta-based protein structure modeling based on the known ADAR2 catalytic domain structure, and revealed an RNA contacting loop structure (5' binding loop fold), involving C1081, C1082, and H1103, which all chelate the newly found Zn ion. The study also revealed a possible interaction between K999 and Y1208, stabilizing the 5' binding loop fold. The importance of the loop structure is further emphasized by the presence of Aicardi-Goutières Syndrome (AGS)-associated mutation K999N, which perhaps disrupts this critical loop fold. These new findings on the ADAR1 catalytic domain structure are very informative and are likely to appeal to general readers of Nature Communication. However, there are several issues, which must be addressed.

Major Concerns:

The authors did not provide sufficient explanation for their interpretation of ICP-MS results (Figure 5).

1) They obtained Zinc/protein mole ratios of 2.67 and 1.07 for ADAR1d and ADAR2d, respectively, and concluded that ADAR1d contains two Zn ions (Figure 5a).

2) The ICP-MS result 2.67 seems to be more reasonably rounded up to 3.0 instead of down to 2.0, and I do not understand the reasoning behind the authors decision to report two, instead of three, Zn ions bound to ADAR1d.

3) In fact, mutagenic analysis of 9 cysteines indicated that not only C1081 and C1082 but also C909 and C1169 are functionally important (Figure 2), perhaps indicating that they are involved in binding to a third Zn ion (?).

Our response: These three points all relate to the zinc:protein stoichiometry and the interpretation of our original ICP-MS data. The three points are addressed together in this response. Since the submission of our original manuscript, we have substantially improved our hADAR1d purification methods and yields. This has allowed us to submit proteins samples at higher concentrations for hADAR1d than previously possible, equaling the concentration of the samples submitted for ADAR2. With this improved sample preparation, ICP-MS shows zinc/protein mole ratios of 1.93 and 0.90 for hADAR1d and hADAR2d, respectively. This supports a stoichiometry of two zinc ions for each hADAR1d protein (Fig. 5a). In addition, we conducted protein native mass spectrometry analysis of hADAR1d and further confirmed the mass corresponding to two zinc ions, one organic cofactor (IP₆), and one protein monomer (Fig. 5b).

Because of difficulty of solubilizing ADAR1d mutant proteins, they prepared all mutant proteins as maltose-binding protein fusion proteins (MBP-ADAR1d) to be used for ICP-MS analysis and in vitro editing assay.

1) They obtained Zinc/protein mole ratio of 1.24 for MBP-ADAR1d-WT, indicating that only one Zn ion (presumably catalytic center Zn) is retained in this fusion protein. Yet, MBP-ADAR1d-WT exhibited full editing activity on Gli1 RNA in vitro ((Figure 6b). The result indicates that the second (or even third) Zn ion is not essential.

Our response: As stated in the manuscript, we believe MBP domain is capable of solubilizing misfolded and demetallated hADAR1d, likely leading to a mixture of demetallated protein and fully metallated protein.

2) Furthermore, Zinc/protein mole ratios of 0.251 and 0.176 were obtained for MBP-ADAR1d-C1082E and ADAR1d-C1082E, respectively. One interpretation of these results is that both Zn ions (catalytic center and second one) are lost in these mutants. Perhaps the loop structure consisting of the second Zn and C1081, C1082, and H1103 somehow affects the stability of the catalytic center structure and retention of the catalytic center Zn ion. Loss of the catalytic center Zn ion for these two mutants could easily explain their total loss of editing activities in vitro (Figure 6b) and in vivo (Figure 6c).

Our response: It is highly possible that the second zinc site plays an essential role in maintaining the global protein structure. Indeed, the corresponding site in ADAR2 has residues K, Q, and Y that engage

in hydrogen bonding and hydrophobic contacts that appear to stabilize structure of the protein. This point is discussed in the manuscript.

The authors should employ an alternate strategy, other than direct ICP-MS to confirm the exact number of Zn ions bound to ADAR1d. They should acquire the Zn-binding data needed for a classical Scatchard plot, which would provide not only the stoichiometry of Zn binding, but also affinity constants and any significant interaction among the 2 (or 3) binding sites. The binding data for this plot could be obtained using ICP-MS to observe the competition between bound "natural" zinc and one of its stable isotopes when the protein is titrated with a solution of the latter. Or perhaps isothermal titration calorimetry might be used, though this involves the risk of denaturing the protein when it is stripped of zinc. In any case some additional support is needed for the authors' claimed stoichiometry.

Our response: We agree with this reviewer that data from an additional experiment would be helpful in supporting our conclusion of a 2:1 (zinc:protein) stoichiometry for hADAR1d. Therefore, we carried out native protein mass spectrometry and confirmed the mass corresponding to two zinc ions, an organic cofactor (IP₆), and a protein monomer (Fig. 5b).

Reviewer #3 (Remarks to the Author):

Review of Park et al.,

In this manuscript the catalytic domain of ADAR1 is investigated. The authors use a mutagenesis approach to interrogate residues located in the 5' binding region of the catalytic domain to predict a zinc-binding region. Zinc-binding is further verified by ICP-MS and mutants in the second zinc coordinating site are being tested for activity in a collection of editing substrates. Modeling is used to predict the structure of the 5' interacting region and to overlay the catalytic domains of ADAR1 and ADAR2.

The manuscript is well written and the flow of experiments presented appears logic, convincing, and complete.

I only have a few comments that should be addressed:

**) Why did the authors only focus on C1082 when testing for their impact on catalytic activity (Fig 6)? Were other mutants tested as well?*

Our response: We focused our activity assay on C1082 mutants because our mutagenesis study suggested that mutation at C1082 position retains some activity. Focusing on this mutation allowed us to confirm the effect of specific mutations at C1082 position on both the metal content and the activity. However, our fluorescence activity assay of H1103X (Fig. 4d) shows that histidine to cysteine mutation at 1103 position retains WT-like activity whereas mutations to other amino acids (A, Q, F, and S) do not. Additionally, we observed a decrease in activity with the H988D mutant in our previous Sat-FACS-seq study (Supplementary Fig. 4).

**) an extended alignment of the ADAR1 catalytic domain would help to get a better idea of the structural elements involved (alpha helices, beta sheets, 5' Interacting region, catalytic core). Such an alignment could be provided in the supplements and would allow the reader to get better oriented.*

Our response: We agree that such an alignment is helpful and have added it as a new figure in the Supporting Information.

**) would it be possible to replace the 5' interacting region of ADAR1 with that of ADAR2? (I realize that this may be far-fetched but if a functional hybrid enzyme can be generated it would allow many predictions to be tested).*

Our response: The 5' binding loop swapping experiment was carried out and published previously from our lab (Wang, Y., Park, S., Beal, P. A, *Biochemistry*, 2018). This study showed that replacing the 5' binding loop of hADAR1d with that of hADAR2d shifts the substrate preference from hADAR1d to hADAR2d, indicating the importance of the 5' binding loop in editing specificity.

REVIEWERS' COMMENTS:

Reviewer #1 (Remarks to the Author):

I am satisfied that the authors have fully addressed the concerns raised.

Reviewer #2 (Remarks to the Author):

The authors responded to comments of all three reviewers, and the manuscript with new data and figures is now significantly improved. This study certainly deserves publication in Nature Communications.

Reviewer #3 (Remarks to the Author):

In this revised version of the manuscript the authors report additional experiments to determine the position of the 2nd Zn-binding site. Indeed, mutational analysis supports the idea of the second Zn-binding region.

The authors have also correctly referred to their previously reported domain swapping experiments and have included an overview on the domain details of ADAR1.

Overall, the authors have fully responded to all points raised.